# Why Are Perivascular Spaces Important?

**DOI:** 10.3390/medicina59050917

**Published:** 2023-05-10

**Authors:** Tatyana Shulyatnikova, Melvin R. Hayden

**Affiliations:** 1Department of Pathological Anatomy and Forensic Medicine, Zaporizhzhia State Medical University, Mayakovsky Avenue, 26, 69035 Zaporizhzhia, Ukraine; shulyatnikova.tv@gmail.com; 2Department of Internal Medicine, Endocrinology Diabetes and Metabolism, Diabetes and Cardiovascular Disease Center, University of Missouri School of Medicine, One Hospital Drive, Columbia, MO 65211, USA

**Keywords:** brain MRI, CADISIL, capillary rarefaction, cerebral small vessel disease, enlarged perivascular spaces, glymphatic system, impaired cognition, lacunes, neurodegeneration, white matter hyperintensities

## Abstract

Perivascular spaces (PVS) and their enlargement (EPVS) have been gaining interest as EPVS can be visualized non-invasively by magnetic resonance imaging (MRI) when viewing T-2-weighted images. EPVS are most commonly observed in the regions of the basal ganglia and the centrum semiovale; however, they have also been identified in the frontal cortex and hippocampal regions. EPVS are known to be increased in aging and hypertension, and are considered to be a biomarker of cerebral small vessel disease (SVD). Interest in EPVS has been significantly increased because these PVS are now considered to be an essential conduit necessary for the glymphatic pathway to provide the necessary efflux of metabolic waste. Metabolic waste includes misfolded proteins of amyloid beta and tau that are known to accumulate in late-onset Alzheimer’s disease (LOAD) within the interstitial fluid that is delivered to the subarachnoid space and eventually the cerebral spinal fluid (CSF). The CSF acts as a sink for accumulating neurotoxicities and allows clinical screening to potentially detect if LOAD may be developing early on in its clinical progression via spinal fluid examination. EPVS are thought to occur by obstruction of the PVS that associates with excessive neuroinflammation, oxidative stress, and vascular stiffening that impairs flow due to a dampening of the arterial and arteriolar pulsatility that aids in the convective flow of the metabolic debris within the glymphatic effluxing system. Additionally, increased EPVS has also been associated with Parkinson’s disease and non-age-related multiple sclerosis (MS).

## 1. Introduction

Perivascular spaces (PVS) (also known as Virchow–Robin spaces (VRS)) are fluid-filled spaces that ensheath the pial arteries, pre-capillary arterioles that reside in the subarachnoid space (SAS) and dive deep into the brain and the effluxing pial post-capillary venules and veins as they exit the brain parenchyma back to the SAS and eventually to the cerebrospinal fluid (CSF). These vessels carry with them a thin continuous lining of squamous pia matter cells as they penetrate the cortical and subcortical structures of the brain. Thus, these spaces are bounded by the inner mural cell basement membranes of the vessels (brain endothelial cell(s) (BEC) and pericyte(s) (Pc)) and are lined by the outer pia cell lining and the astrocyte end-feet (ACef) basal lamina, which appear to be fused of the parenchymal side of the PVS. The arterial PVS are the primary delivery conduit for the CSF to admix and promote movement within the interstitial fluid (ISF). The ACef basal lamina-lined post-capillary venules and veins are important for the efflux of metabolic waste via the PVS–glymphatic pathway from the interstitial ISF to the CSF sink (Figure 1, Figure 2, Figure 3 and Figure 4) [1,2,3,4]. According to TEM studies by Zhang et al., the post-capillary venule and veins are not ensheathed by the pia matter lining as in the penetrating arteries and pre-capillary arterioles [2].

PVS are considered to be enlarged PVS (EPVS) once they are large enough to be identified on T-2-weighted magnetic resonance imaging (MRI) and are usually greater than one micrometer (averaging between 1 and 3 μm) [5]. Because EPVS can now be visualized and identified by MRI, they are currently known to be clinically relevant. In addition to visualizing PVS and EPVS by MRI, PVS and EPVS may also be identified by transmission electron micrographic (TEM) studies (Figure 4) [6].

Furthermore, recent studies reported that EPVS are associated with a genetic predisposition to Alzheimer’s disease (AD), providing evidence that the biological pathways affecting AD may also influence EPVS [7]. In this perspective, microscopy plays a crucial role, being a fundamental tool in the clinical setting. Light microscopy is an essential tool for describing the significant morphological changes in tissues and TEM is a powerful instrument, which provides ultrastructural evidence of tissue remodeling, to understand the physiological and pathological dynamics of PVS development and remodeling [8].

Moreover, EPVS have been increasingly recognized as important structural remodeling changes in various neuropathologies [9]. They commonly associate with advancing age (PVS are formed postnatally and are known to become enlarged during the aging process) [8], hypertension (HTN), lacunes, microbleeds, intracerebral hemorrhages, cerebrocardiovascular disease (CCVD) with transient ischemic episodes (TIA) and stroke, SVD, cerebral autosomal dominant arteriopathy with subcortical infarcts and leukoencephalopathy (CADISIL), cerebral amyloid angiopathy (CAA), obesity and metabolic syndrome, cerebral small vessel disease (SVD), white matter hyperintensities (WMH), late-onset Alzheimer’s disease (LOAD), sporadic Parkinson’s disease (PD), and non-age-related multiple sclerosis [4,5,10,11,12,13,14,15,16,17]. Despite this long list of numerous associations, more studies are required to determine the possible pathophysiological involvement in regard to cerebrocardiovascular, neuroinflammatory, and neurodegenerative disorders [9,10]. Thus, there is great interest in the pathobiology of neurovascular, neuroinflammatory, and neurodegenerative diseases in regard to EPVS.

EPVS are paired structures and are known associate with WMH and lacunes and have been determined to consist of at least three major types based on their location. Type I PVS appear alongside lenticulostriate arteries to enter the BG and are known to have a double coating of pia matter. Type II PVS appear alongside the perforating medullary arteries as they enter cortical gray matter that extends into the white matter CSO. Type III PVS appear alongside the penetrating branches of the collicular and accessory collicular arteries that enter the midbrain [16].

Most neurological diseases are described as being multifactorial and involve an overlapping contribution of at least three dysfunctional mechanisms: neurovascular, neuroinflammatory, and neurodegenerative. (1) Aberrant neurovascular mechanisms include neurovascular unit (NVU) uncoupling; blood–brain barrier (BBB) dysfunction or disruption, which includes tight and adherens junction (TJ/AJ) dysfunction, attenuation, and/or loss; increased BEC transcytosis; BEC activation and dysfunction; and increased capillary rarefaction with decreased capillary density. (2) Aberrant neuroinflammatory mechanisms include the promotion of oxidative stress, and oxidative stress promotes ongoing inflammation that contributes to advanced glycation end products/receptors for the interaction of advanced glycation end products (AGE/RAGE) to further increase oxidative stress. (3) Increased neurodegenerative mechanisms are exemplified by a brain injury early on that is recapitulated, over and over, by the response to injury wound healing mechanism that is genetically programmed during embryonic development. Initially, the response to injury wound healing mechanisms is protective; however, if these mechanisms are sustained over prolonged periods of time, they promote neuropathology. Trolli et al. have suggested that the PVS be considered as a unit termed the perivascular unit (PVU) [16]. The substrate cells for this unit consist of brain endothelial cell(s) (BEC), pericyte(s) (Pc), pia matter cells, and astrocyte(s) (AC) and their end-feet (ACef) and their basal lamina. Further, these authors propose that the PVU serves as a crossroad for the interaction of neurovascular, neuroimmune, and neurodegenerative mechanisms of brain injury and response to injury wound healing mechanisms [17,18,19].

The brain does not have a classic lymphatic drainage system that is present in the peripheral vascular system; however, a glymphatic system that utilizes the PVS as a waste efflux conduit has been recently described by many and serves to remove toxic waste from the interstitium of the brain [1]. While the glymphatic system is of great importance and not to be overlooked, this review does not lend itself to going into detail regarding the glymphatic system, but for those who are more interested, please see Reference [1].

This brief narrative review also discusses the EPVS association with basal ganglia (BG) and centrum semiovale (CSO), lacunes, WMH, and SVD in Section 2; brain endothelial cell activation and dysfunction (BEC*act/dys*) and BEC glycocalyx (ecGCx) shedding associated with EPVS and SVD in Section 3; and metabolic syndrome (MetS), SVD, and PVS in Section 4.

## 2. EPVS Association with Basal Ganglia (BG) and Centrum Semiovale (CSO), SVD, Lacunes, and WMH

It is no wonder that EPVS associate with SVD, WMH, and lacunes, since structurally, the PVS ensheath the microvessels that penetrate and supply the deep myelinated white matter, including the paired BG and CSO, and drain the waste products from the interstitium of the neuropil (Figure 1, Figure 2 and Figure 3).

The BG are a paired grouping of subcortical nuclei (caudate nucleus, putamen, and globus pallidus with input, output, and intrinsic pathways) structures linked to the thalamus at the base of the brain and involved in the coordination of movement and motor control in addition to learning, executive functions, behaviors, and emotions [20]. The paired CSO with semi-oval shaped white matter tracts (projection, commissural, and association pathways) are located superior to the lateral ventricles and corpus callosum that are present in each of the cerebral hemispheres and adjacent to the overlying gray matter cerebral cortex [21].

Indeed, SVD with decreased cerebral blood flow (CBF) may be involved in a bidirectional relationship between the development of EPVS, WMH, and lacunes. Importantly, EPVS, WMH, and lacunes are now thought to strongly associate with SVD.

EPVS, WMH, and lacunes are now considered as biomarkers of the development and progression of SVD and clinical complications of TIAs and stroke (ischemic and hemorrhagic) with increased morbidity and mortality [4,9,10,11,13,14,15,17,18,22,23,24].

From a clinical standpoint, SVD presents and is associated with lacunar strokes that are responsible for at least 20% of ischemic strokes and represent a major cause of vascular cognitive impairment. Additionally, EPVS are known to be a biomarker and a feature of both SVD and vascular dementia (VaD), which are known to be associated with lacunar stroke as well as WMH [4,14,25,26,27]. Therefore, it is important to distinguish between lacunes, EPVS, and WMH as it pertains to SVD and strokes (Figure 5 and Figure 6).

The endothelium of microvessels within cortical gray matter and especially the subcortical white matter plays an important role in the formation of lacunes, EPVS, and WMH in the development of SVD (Figure 7) [26,27,28,29,30,31].

Notably, there may exist an evolutionary spectrum wherein EPVS progress over time to result in SVD, neuroinflammation, impaired cognition, and neurodegeneration (Figure 8).

## 3. BEC*act/dys* and BEC Glycocalyx (ecGCx) Shedding Associate with EPVS, and SVD

BEC*act/dys* is important for the development of EPVS and subsequent SVD. BEC*act/dys* includes BEC activation that associates with vascular BEC inflammation–neuroinflammation and is characterized by increased expression of cell-surface adhesion molecules that include intercellular adhesion molecule 1 (ICAM-1), vascular cell adhesion molecule 1 (VCAM-1), and endothelial–leukocyte adhesion molecule 1 (E-selectin). BEC dysfunction is characterized by decreased synthesis, release, and/or activity of endothelium-derived nitric oxide, which results in decreased bioavailable nitric oxide (NO) (Figure 9) [4,6,31,32,33].

In the past decade, the authors have been able to identify multiple TEM remodeling changes associated with activated BECs that may contribute to EPVS and subsequent SVD, as outlined in Figure 10 [6].

Indeed, there are multiple toxicities that are known to activate BECs, which include infections (viral, bacterial, and parasitic), elevated homocysteine, angiotensin II, redox stress, glucotoxicity, lipotoxicity, modified low-density lipoprotein (LDL) cholesterol, and hemodynamic stressors (hypertension) [33]. Importantly, these toxicities are known to accelerate both atherosclerosis and arteriolosclerosis that associate with SVD [34].

Possible mechanisms for the development of EPVS and subsequent SVD include (1) increased proteins and fluid coming into PVS due to increased permeability of BBB due to dysfunctional and/or disrupted TJ/AJ with paracellular influx or by the transcytotic route via increased transcytosis of both micro- and macropinocytotic vesicles of the activated BECs (Figure 11) [35]; (2) decreased fluid outflow from PVS due to impaired or dysfunctional astrocyte end-feet due to detachment or separation from the NVU with decreased fluid uptake by the aquaporin-4 (AQP4) water channels allowing fluid to accumulate in the PVS; (3) obstruction of PVS via excessive PVS inflammation, oxidative stress, and activation of matrix metalloproteinases resulting in excessive extracellular debris, which results in decreased PVS fluid flow with PVS enlargement or dilation with the stagnation of waste removal mechanisms; (4) arteriole or venule stiffening that is associated with decreased vascular pulsatility that results in decreased fluid flow within the PVS that contributes to enlargement; and (5) atrophy or loss of surrounding neurons and their axons allowing the PVS to expand [4,9,13,14,15,16,17,18,19,20,21,22,23,24,25,26,27,28,29].

EPVS fluid flow and clearance are impaired in models of stroke, multiple infarct dementias, diabetes, traumatic brain injury, and CADASIL [4]. Narrowing of the PVS and EPVS with impaired fluid waste transport has also been demonstrated in models of migraine that are present in individuals with CADASIL. Dysfunction of the glymphatic system contributes to the accumulation of toxic waste and interstitial edema, and instigates pathological remodeling changes that affect the impact of brain health and accelerated brain aging. Thus, EPVS may act not only as a biomarker of SVD but also imply impaired fluid transport and waste clearance of the EPVS and the glymphatic pathway. The brain does not have a classic lymphatic system that is present in the peripheral vascular system; however, a glymphatic system that utilizes the PVS has been recently described by many researchers and serves to remove toxic waste from the interstitium of the brain [4,37,38,39]. Further, these EPVS and impaired efflux of PVS and glymphatic pathway may be bidirectional with one aggravating the other [4,37,38,39].

In health, the brain ecGCx is vasculoprotective and plays a significant role in vascular integrity and homeostasis [40,41]. It may be considered the first barrier of a tripartite barrier, which includes (i) ecGCx, (ii) BEC and its BM, and (iii) BEC BM and the astrocyte end-feet of the extravascular compartment of the NVU (Figure 12) [42].

Dysfunction, attenuation, and/or loss of the ecGCx results in disruption of BBB TJ/AJ integrity with subsequently increased permeability and thus associates with increased fluid being transferred into the PVS as discussed previously in regard to BEC*act/dys* (Figure 11 and Figure 12) [28,29,40]. Importantly, there is strong emerging evidence that there exists a bidirectional role between the accumulation of BEC aberrant mitochondria (aMt) and dysfunction or loss-shedding of the BEC ecGCx (Figure 13) [30,43,44,45,46].

While more research may be necessary in order to confirm this bidirectional relationship it nevertheless remains an intriguing association and presents an emerging opportunity to further unlock some of the mysteries associated with BEC aMt and increased mtROS, shedding of BEC ecGCx, and BEC*act/dys* in the development BBB dysfunction, disruption, EPVS, and SVD.

## 4. Metabolic Syndrome (MetS), Perivascular Spaces (PVS), and Cerebral Small Vessel Disease (SVD)

MetS is known to be a cluster of multiple risk factors and variables that are associated with an increased risk of the development of atherosclerotic cerebrocardiovascular disease (CCVD) and type 2 diabetes mellitus (T2DM) (Figure 14) [6,47,48].

There are four core features, namely hyperlipidemia, hyperinsulinemia, hypertension, and hyperglycemia [6,47,49]. MetS, EPVS, and SVD are each known to be associated with aging [5,6,15,16,32]. Obesity and MetS are increasing globally due to an aging population, urbanization, a sedentary lifestyle, and increased caloric diets high in fat, sucrose, and glucose [6,47,48,49]. Currently, we have one of the oldest global populations in our history [49]; therefore, it is not surprising that we are observing a global increase in not only MetS but also in EPVS and SVD [4,5,6,14,15]. Further, MetS is associated with the development of EPVS and SVD [22,50,51,52].

Recently, it has been demonstrated that MetS is associated with capillary rarefaction that is accentuated when there is co-existent decreased NO bioavailability [53,54]. Capillary rarefaction (loss of capillaries) is a condition wherein there is a decrease in small vessel capillary density that occurs in the brain. This decrease in the number of capillaries may have regional variations with certain disease processes and vary between different organs. Examining Figure 14, note the intricate relationship between visceral obesity (VAT), MetS, and decreased bioavailable NO [54], also associated with co-existing BEC*act/dys* previously emphasized in Figure 11 and Figure 14. Additionally, advancing age contributes to obesity, MetS, and decreased bioavailable NO, similar to how aging also contributes to EPVS and SVD [54].

Visceral obesity, MetS, decreased bioavailable NO, and advancing age contribute to cerebromicrovascular–capillary rarefaction (Figure 14) [53,54,55,56]. It is of interest to note that during the process of capillary rarefaction with capillary loss, an empty space will develop within the PVS that ensheath the pre-capillary arterioles and post-capillary venules. This loss of true capillaries, pre-capillary arterioles, and post-capillary venules will allow for an increase in the total volume of the fluid-filled spaces within the PVS and may contribute to EPVS (Figure 15) [53,54,55,56].

Since obesity and MetS are associated with both capillary rarefaction and EPVS, it is entirely possible that the above mechanistic hypothesis may help to explain the increased fluid in the PVS with increased volume of fluid once there is capillary loss, as illustrated in Figure 15. While this mechanistic hypothesis is possible, there will need to be more research carried out to further support this mechanistic process and the relationship between capillary and the development of very early EPVS.

Indeed, MetS is associated with cognitive impairment and structural remodeling, and potential explanations include IR, LR, oxidative stress, neuroinflammation, aberrant lipid metabolism, and neurodegeneration [31,57]. Cerebrovascular reactivity impairment and capillary rarefaction are associated with NVU uncoupling. NVU uncoupling in MetS results in chronic regional hypoxia and is associated with impaired cognitive function and over time may result in neurodegeneration [57].

## 5. Conclusions

In older community-dwelling individuals free of clinical dementia and stroke, SVD biomarkers, including EPVS, WMH, and lacunes, are related to worse cognitive performance [57]. Indeed, previous studies have shown that EPVS are associated with worse executive function and information processing in healthy older adults [57] and are significantly more prevalent in those with mild cognitive impairment (MCI) as compared to age-matched control subjects without MCI [58]. These findings certainly suggest that EPVS may be an early remodeling change in the development of SVD and impaired cognition [59,60].

There are multiple possible mechanisms for cognitive damage, impairment, and remodeling caused by SVD, which include the following four mechanisms: (1) The presence of EPVS, WMH, and lacunes (biomarkers for SVD) can disrupt the white matter brain network integrity and normal cognitive function because of a loss of cortical integration and structural disconnection of white matter cortical tracts and result in cortical atrophy; (2) BBB dysfunction and/or disruption of the NVU leads to vascular damage in SVD as a result of decreased CBF and impaired vasodilation in response to neuronal activity not only in the hippocampus but also the cortical regions and the vulnerable white matter regions; (3) BBB dysfunction and/or disruption allows increased NVU permeability with leakage of fluids and plasma protein neurotoxins into the PVS, and promotes resident macrophage reactivity in PVS with associated neuroinflammation that could impair the glymphatic system efflux function. The leakage of the PVS contents into the interstitial spaces would then be capable of promoting neuroinflammation and the accumulation of toxic proteins which then may undergo aggregation and deposition, supporting a two-step process for neuroinflammation [58]. (4) Increased SVD, as indicated by increased biomarkers (EPVS, WMH, and lacunes), contributes to impaired interstitial fluid efflux via the glymphatic pathway with impaired clearance of neurotoxic protein elimination [61].

These above four possible mechanisms are important because neurovascular and neurodegenerative mechanisms co-occur as mixed and co-occurrence dementias in age-related neurodegenerative diseases such as LOAD and sporadic PD [19]. Thus, EPVS as biomarkers of SVD become increasingly important in addition to WMH, cerebral microbleeds, lacunes, and SVD [62]. Additionally, EPVS, as identified on MRI, are associated with microvascular WMH, lacunes, SVD, advancing age, impaired efflux by the glymphatic system, and numerous clinical neurologic diseases, as discussed in this brief narrative.

This narrative review parallels many of the referenced published papers; however, the authors have also utilized TEM images regarding PVS and EPVS to allow for a better understanding of the associated ultrastructure remodeling. Additionally, the authors have provided many schematic illustrations to aid in the understanding of why PVS and EPVS are important.

As our global aging population continues to grow, EPVS are becoming an increasingly important structural abnormal finding, since they also relate to clinical extracranial atherosclerosis [6,62], neurovascular cerebromacrovascular and cerebromicrovascular disease, and age-related neurodegenerative diseases such as LOAD and sporadic PD. PVS are important for many different reasons, and at least eight core reasons are suggested in Figure 16.

In addition to these eight core reasons for why PVS are important, more reasons will undoubtedly be revealed as research continues in this ongoing and exciting field of study.

In regard to future directions, the use of artificial intelligence and deep machine learning algorithms may help to improve our knowledge of the relationship between EPVS, SVD, and impaired cognition in large, combined cohorts. These evolving, unbiased methods may help to provide more reliable, clinically meaningful results. Furthermore, these results would not be confounded by previous observer hand-counting methods that have been used in the past, in addition to decreasing the amount of time involved to generate large datasets of information [63].

## Figures and Tables

**Figure 1 medicina-59-00917-f001:**
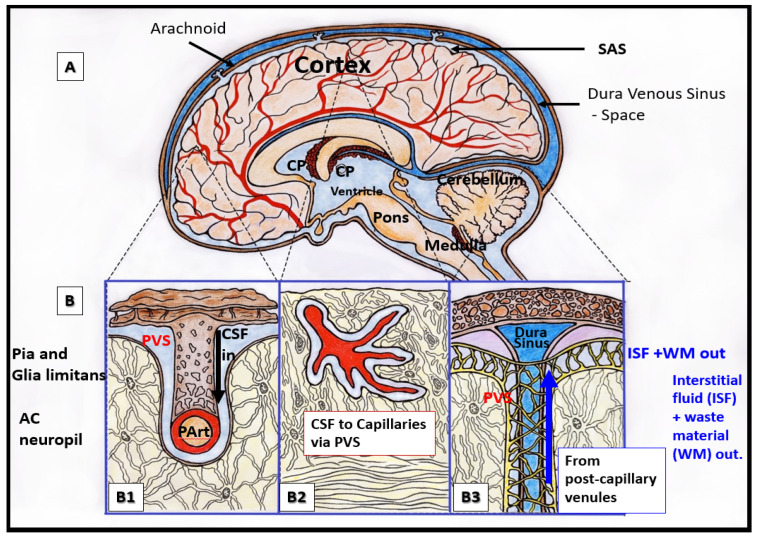
Relationship of perivascular spaces PVS to the whole brain. (**A**) illustrates the structurally labeled whole brain with demarcations of specific regions (dashed lines). (**B**) depicts the pia artery within the subarachnoid space (SAS) that penetrates the deeper brain structures in a perpendicular manner with adjacent PVS with blue coloration (**B1**) and horizontally–diagonally (**B2**), wherein the perivascular spaces (PVS) allow for the influx (black arrow) of the cerebrospinal fluid (CSF) to the parenchymal interstitial fluid space (ISF) via the arteriolar PVS. Panel (**B3**) depicts the efflux (blue arrow) of the interstitial fluid metabolic waste material (WM) of the pial venular PVS to the pial vein PVS that enter the subarachnoid space to eventually drain into the dural venous sinus space.

**Figure 2 medicina-59-00917-f002:**
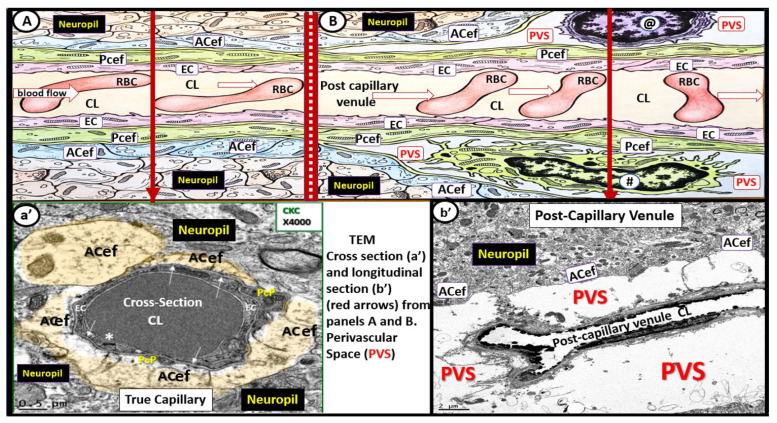
Illustration of the transition from a true capillary to a post-capillary venule with accompanying transition electron microscopy (TEM) images. (**A**) illustrates a true capillary without perivascular spaces (PVS). (**B**) illustrates a post-capillary venule with PVS, which contains a resident macrophage (#) and lymphocyte (@). Note the direction of blood flow within the capillary lumen (CL) (open red arrows from left to right in these images (**A**,**B**)). Note the EC tight and adherens junction (*). Additionally, note that the closed red arrows in (**A**,**B**) point to the corresponding TEMs in (**a’**,**b’**). (**a’**) demonstrates a cross-section electron micrograph of a true capillary and how the astrocyte end-feet (ACef) directly abut the basement membrane of the mural endothelial cell(s) (ECs) and pericyte end-feet process (Pcef PcP). (**b’**) depicts a longitudinal section of the post-capillary venule with prominent PVS. *RBC* = *red blood cell*.

**Figure 3 medicina-59-00917-f003:**
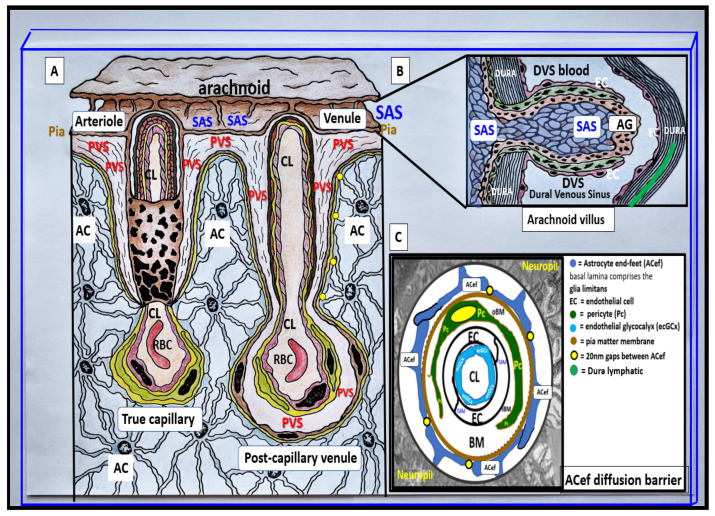
Pial pre-capillary influx arteriole, post-capillary efflux venules, and perivascular spaces (PVS). (**A**) illustrates the PVS bounded by the arteriole and venous endothelial/pericyte basement membranes and the pia matter/astrocyte end-feet basal lamina (glia limitans). The arteriole and precapillary PVS are responsible for the influx of cerebrospinal fluid (CSF) from the cerebrospinal fluid (CSF) and subarachnoid space (SAS) to the parenchymal interstitial fluid (ISF) space; some studies have demonstrated a retrograde efflux of ISF and CSF against the flow to the SAS, while the post-capillary venule and veins are responsible for the efflux of the ISF and the admixed CSF, and metabolic waste to the SAS and eventually to the dural venous sinus (DVS) via arachnoid granulation(s) (AG). Importantly, note how the pia matter membrane covering abruptly disappears at the level of the true capillary that is now covered by only the astrocyte end-feet (ACef) (glia limitans) on the outer capillary neurovascular unit. Additionally, note that the pia matter layer is thought to be not present in the post-capillary venules and veins [2]. (**B**) illustrates an arachnoid villus and its AG for the exchange of ISF, CSF, and metabolic waste with the DVS blood and the dural lymphatics (cyan). The metabolic waste is also known to drain along cranial nerve sheaths or through the nasal lymphatic system. (**C**) depicts the astrocyte end-feet (ACef) barrier with only a few 20 nm gaps and thus creates the rate-limiting barrier for water and solute exchange. Importantly, the ACef contain the polarized aquaporin-4 (AQP4) water channel, which has also been shown to be important in fluid and solute exchange.

**Figure 4 medicina-59-00917-f004:**
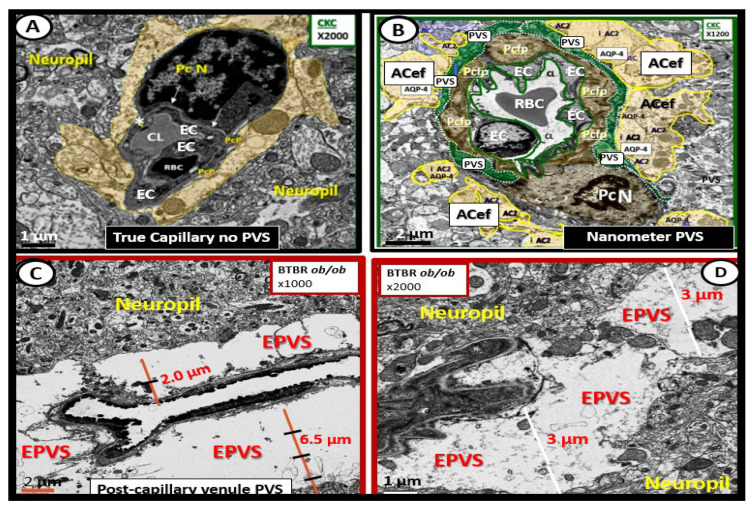
Ultrastructure comparison of perivascular spaces (PVS) and enlarged perivascular spaces (EPVS) utilizing transmission electron microscopy (TEM). (**A**) demonstrates a true capillary in a control C57B6-7 model. Note how the pseudo-colored golden astrocyte end-feet (ACef) tightly abut the basement membrane (BM) of the neurovascular unit (NVU) brain endothelial cell (BEC or EC) that does not have a PVS. (**B**) demonstrates a very small nanometer PVS (pseudo-colored green). Note how the pseudo-colored golden ACef do not tightly abut the combined BEC and pericyte (Pc) BM as in the true capillary in (**A**). However, this nanometer-sized PVS is bounded by the abluminal ACef and pia matter (glia limitans) of this terminal arteriole before it transitions to a true capillary without a PVS. (**C**) depicts a post-capillary venule with an EPVS varying from 2 to 6.5 μm in diameter in the obese diabetic black and tan brachyuric *ob/ob* (BTBR *ob/ob)* transgenic mouse model. (**D**) depicts EPVS with a 3-micrometer diameter space in the BTBR *ob/ob* model. Note how the surrounding ACef now abut the EPVS on its most abluminal boundary in (**C**,**D**) that are bounded by its innermost BEC and Pc basement membrane. Magnifications and scale bars vary and are present in (**A**–**D**). *AQP4* = *aquaporin-4 water channel; CL* = *capillary lumen*; *EC* = *brain endothelial cell*; *Pcfp* = *pericyte foot process or end-feet*; *Pc N* = *pericyte nucleus*; *RBC* = *red blood cell*.

**Figure 5 medicina-59-00917-f005:**
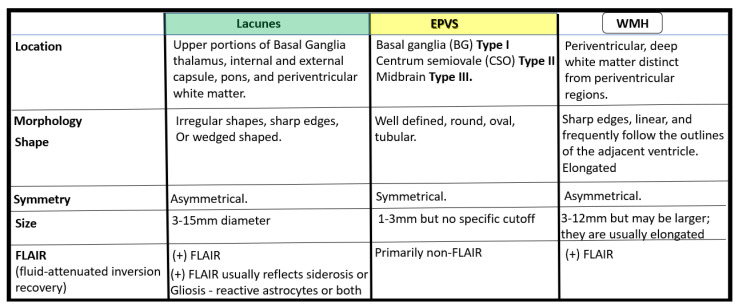
Comparisons between lacunes, enlarged perivascular spaces, and white matter hyperintensities. *mm* = *micrometer*.

**Figure 6 medicina-59-00917-f006:**
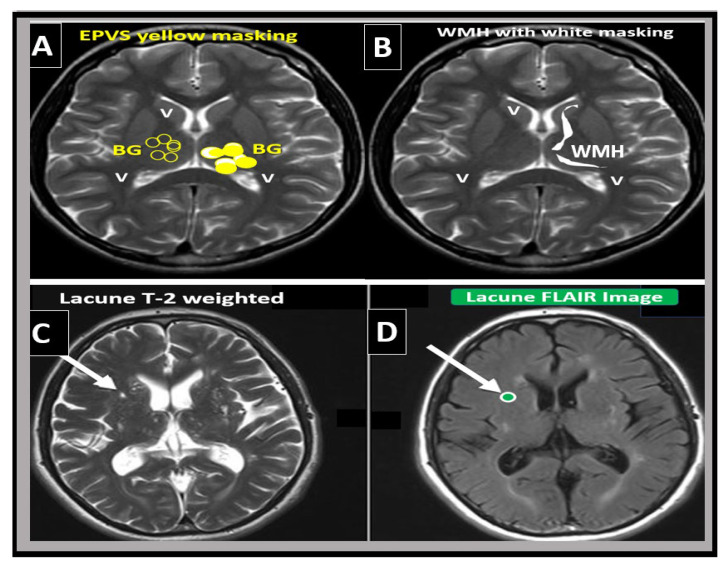
Magnetic resonance imaging (MRI) comparison of enlarged perivascular spaces (EPVS), white matter hyperintensities (WMH), and lacunes. (**A**) depicts EPVS localized to the basal ganglia (symmetrical) with yellow color masking of EPVS (EPVS localized to centrum semiovale not shown). (**B**) depicts WMH localized to the periventricular regions (deep white matter WMH not shown). (**C**) depicts a T-2-weighted lacune (arrow). (**D**) depicts a FLAIR image with cyan color masking. Note the encircling white line to suggest hyperintensity FLAIR. *V* = *ventricle*.

**Figure 7 medicina-59-00917-f007:**
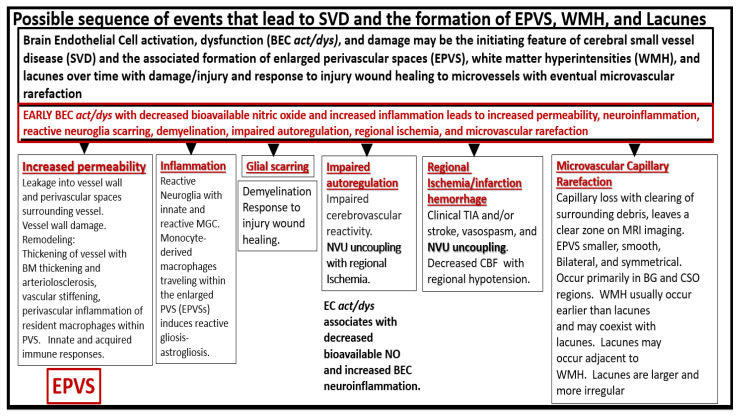
Possible sequence of events that lead to cerebral small vessel disease (SVD) and the formation of enlarged perivascular spaces (EPVS), white matter hyperintensities (WMH), and lacunes. *BEC* = *brain endothelial cell*; *BEC act/dys* = *brain endothelial cell activation and dysfunction*; *BG* = *basal ganglia*; *BM* = *basement membrane*; *CBF* = *cerebral blood flow*; *CSO* = *centrum semiovale*; *MGC* = *microglia cell*; *MRI* = *magnetic resonance imaging*; *NO* = *nitric oxide*; *NVU* = *neurovascular unit*; *TIA* = *transient ischemic attack*.

**Figure 8 medicina-59-00917-f008:**
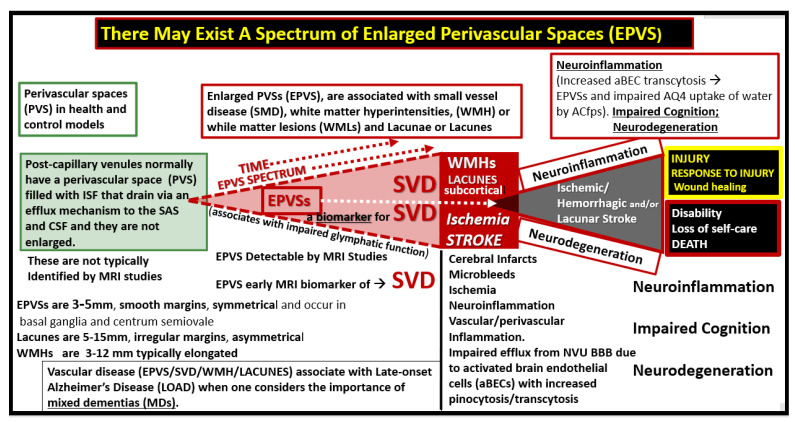
Enlarged perivascular spaces (EPVS) may exist in an evolutionary spectrum over time to result in cerebral small vessel disease (SVD) with neuroinflammation, impaired cognition, and neurodegeneration. *ACfp* = *astrocyte foot processes or end-feet*; *BBB* = *blood–brain barrier*; *CSF* = *cerebrospinal fluid*; *ISF* = *interstitial fluid*; *LOAD* = *late-onset Alzheimer’s disease*; *mm* = *micrometer*; *MRI* = *magnetic resonance imaging*; *NVU* = *neurovascular unit*; *PVS* = *perivascular spaces*; *SAS* = *subarachnoid space*; *WMH* = *white matter hyperintensities*.

**Figure 9 medicina-59-00917-f009:**
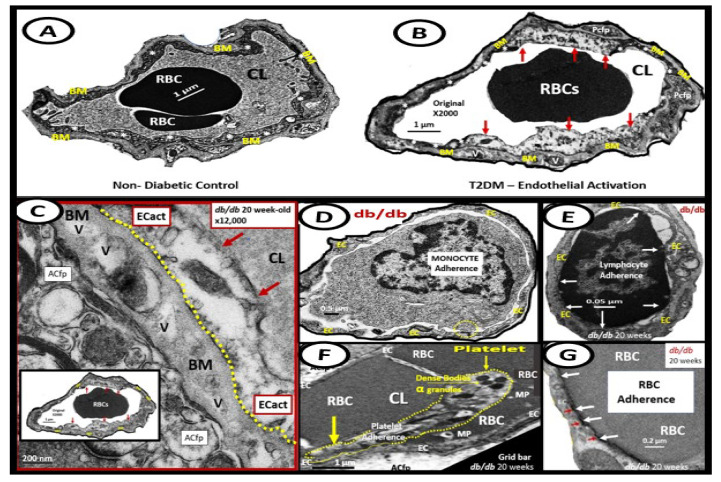
Examples of transmission electron microscopic (TEM) images for endothelial cell activation (EC*act*). (**A**) demonstrates the normal TEM appearance of the brain endothelial cell (BEC). Note the thinness and electron density of this BEC. (**B**) depicts the abrupt appearance of thickened regions of electron-lucent areas (red arrows) of EC*act* compared to the control model in (**A**). (**C**) depicts basement membrane (BM) thickening with increased vacuoles (V) and vesicles (v). (**D**,**E**) depict monocyte (**D**) and lymphocyte (**E**) (white arrows depict cellular adhesion to activated endothelium), platelet (outlined by yellow dashed lines and yellow arrows), and red blood cell (RBC) adhesion (white arrows) (**F**,**G**) to the activated ECs. Original magnification = ×2000; scale bar = 1 μm. Modified with permission CC by 4.0 [6]. Images in (**C**–**G**) were reproduced and modified with permission by CC 4.0 [19]. *ACfp* = *astrocyte foot processes*; *Cl*, *capillary lumen*; *EC* = *brain endothelial cells*; *ECact* = *endothelial cell activation*; *MP* = *microparticle of the platelet*.

**Figure 10 medicina-59-00917-f010:**
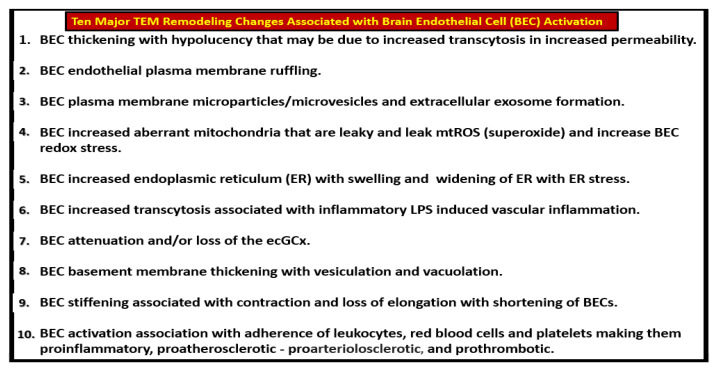
Summary of the observational transmission electron microscopic (TEM) remodeling changes in activated brain endothelial cells in obesity, metabolic syndrome, type 2 diabetes mellitus, and hypertensive rodent models. *MtROS* = *mitochondria reactive oxygen species*.

**Figure 11 medicina-59-00917-f011:**
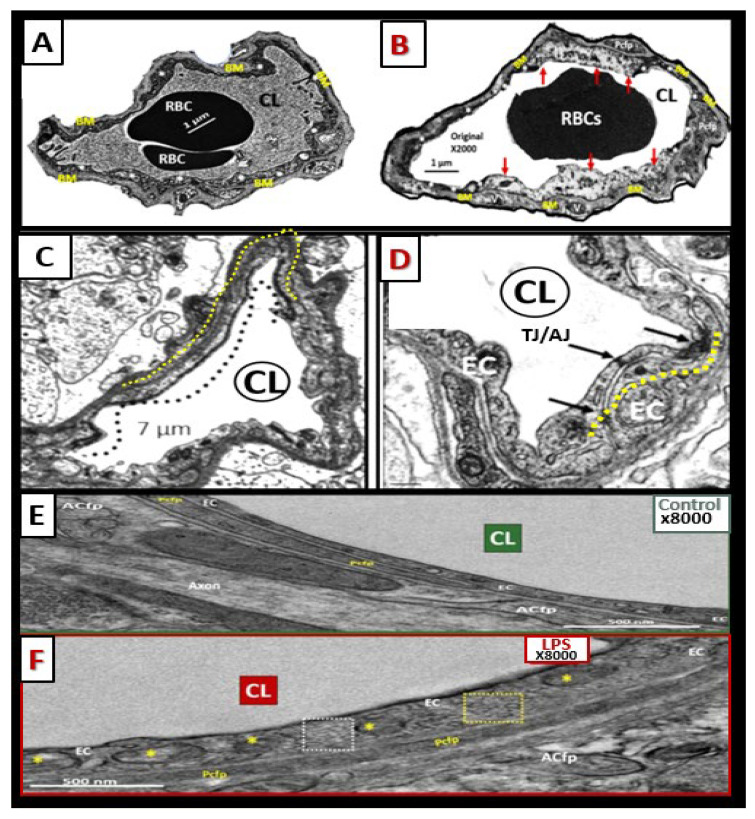
Brain endothelial cell activation, attenuation and/or loss of paracellular tight and adherens junctions, and increased transcytosis may each contribute to enlarged perivascular spaces. (**A**,**C**,**E**) demonstrate the appearance of normal BECs found in control preclinical models. Note the yellow dashed line that parallels the elongated tight and adherens junctions (TJ/AJ) in the control model (**C**). (**B**,**D**,**F**) images depict activation of brain endothelial cells (BECs) as compared to their respective control model images in (**A**,**C**,**E**). (**B**) depicts abrupt swelling and hyperluceny of BECs; (**D**) depicts the disruption, attenuation, and loss of TJ/AJs; and (**F**) depicts increased transcytotic micro- and macropinocytotic vesicles. Each of these aberrant BECs contributes to increased permeability through different mechanisms, i.e., increased permeability via paracellular and transcytotic routes. (**A**,**B**) were provided with permission by CC 4.0 [6]. (**C**,**D**) original images from streptozotocin-induced diabetes in CD-1 male mice at 16 weeks of age. (**E**,**F**) were provided with permission by CC 4.0 [36]. *ACfp or ACef* = *astrocyte foot process or end-feet*; *BM* = *basement membrane*; *CL* = *capillary lumen*; *EC* = *brain endothelial cell(s)*; *LPS* = *lipopolysaccharide*; *Pcfp* = *pericyte foot processes*; *RBC* = *red blood cell*.

**Figure 12 medicina-59-00917-f012:**
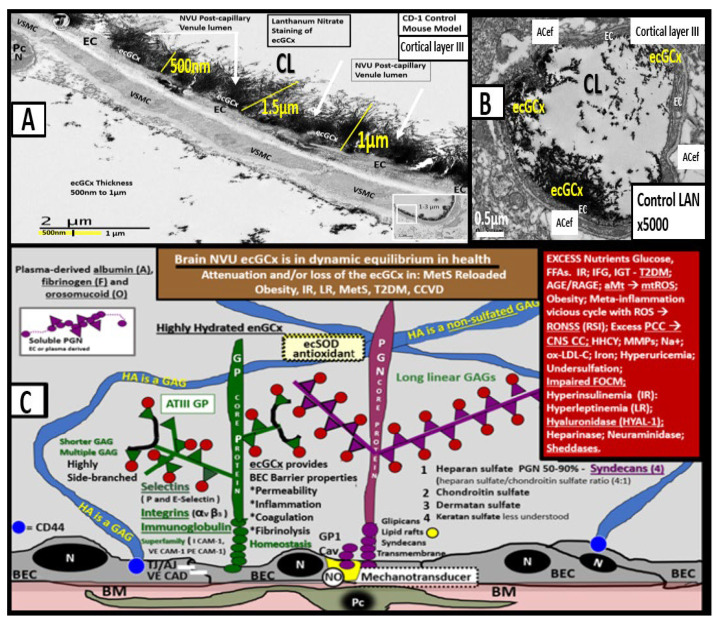
Transmission electron microscopic images of lanthanum nitrate staining of the endothelial glycocalyx (ecGCx) and an illustration of the (ecGCx) as it occurs in the brain neurovascular unit. (**A**) demonstrates the normal lanthanum nitrate (LAN) perfusion fixation staining of the endothelial glycocalyx (ecGCx) (white arrows) in a post-capillary higher-order venule with a single lining of vascular smooth muscle cells (VSMC). This magnified image demonstrates the normal LAN positive staining of the ecGCx in a healthy control male CD-1 mouse brain from the frontal cortical gray matter layer III with the original intact scale bar = 2 μm and yellow scale bar = 1 μm with a 500 nm to 1.5 μm thickness of the ecGCx. Note the boxed-in insert in the lower right-hand corner with a white outline, which is the original image from which the magnified image in (**A**) is derived. (**B**) illustrates another representative image in a true capillary without a PVS that is positive for LAN staining of the ecGCx from cortical layer III frontal gray matter. Note the intense electron-dense staining of lanthanum nitrate of the apical brain endothelial cell(s) (BEC) of the ecGCx such that the structural content of the ecGCx cannot be visualized in either (**A**) or (**B**); however, the components of the ecGCx will be illustrated in (**C**). (**C**) illustrates the various proteoglycans (PGNs) (purple), glycoproteins (GPs) (green), hyaluronan (HA) (blue), glycosaminoglycans (GAGs) (purple and green triangles), and their sulfation sites (red circles). Note the red boxed-in region on the upper-right-hand side of this image that lists the numerous toxicities capable of causing ecGCx dysfunction, attenuation, and/or shedding. This modified and adapted image was provided with permission by CC 4.0 [6,42]. *A* = *albumin*; *AGE/RAGE* = *advanced glycation end; BM* = *basement membrane*; *CAD* = *cadherin*; *CAM* = *cellular adhesion molecule*; *CD44* = *cluster of differentiation 44*; *EC* = *endothelial cell(s)*; *ecSOD* = *extracellular superoxide dismutase*; *F* = *fibrinogen*; *FGF2* = *fibroblast Growth Factor 2*; *FOCM* = *folate-mediated one-carbon metabolism*; *GCx* = *glycocalyx*; *ICAM-1* = *intercellular adhesion molecule*; *Ox LDL* = *oxidized low-density lipoprotein*; *LPL* = *lipoprotein lipase*; *MMPs* = *matrix metalloproteinases*; *N* = *nucleus*; *Na+* = *sodium*; *O* = *orosomucoids*; *Pc* = *vascular mural cell pericyte(s)*; *PECAM-1* = *platelet endothelial cell adhesion molecule-1*; *RONS* = *reactive oxygen species*; *VEC* = *vascular endothelial cell(s)*; *TFPI* = *tissue factor pathway inhibitor*; *TJ/AJ* = *tight and adherens junctions*; *VCAM* = *vascular cell adhesion protein*; *VE CAD* = *vascular endothelial cadherins*; *VEGF* = *vascular endothelial growth factor*; *XOR* = *xanthine oxioreductase*.

**Figure 13 medicina-59-00917-f013:**
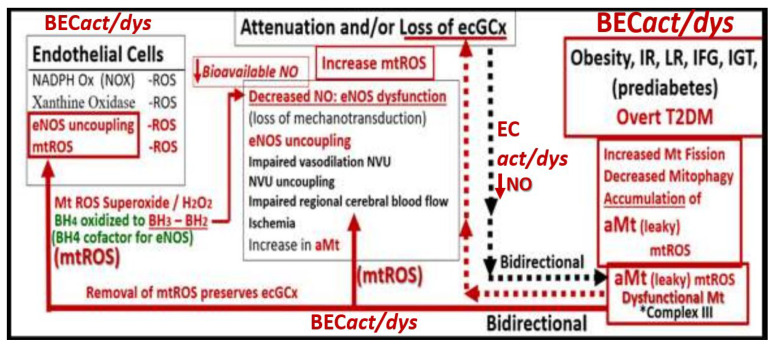
This image illustrates a bidirectional relationship between brain endothelial cell activation/dysfunction (BEC*act/dys*), aberrant mitochondria (aMt) and dysfunction, attenuation, and/or shedding of the brain endothelial glycocalyx (BEC ecGCx). BEC*act/dys* further illustrates the role of BEC oxidative stress and reactive oxygen species (ROS) and specifically mitochondrial ROS (mtROS) (left-hand box). This schematic also shows that obesity, insulin resistance (IR), leptin resistance (LR), impaired fasting glucose (IFG), impaired glucose tolerance (IGT), lipotoxicity (including modified low-density lipoprotein-cholesterol (mod-LDL-C)), and overt type 2 diabetes mellitus (T2DM) are related to increased Mt fission, decreased mitophagy, and the subsequent accumulation of leaky aMt that leak mtROS (right-hand box). Leaky aMt may be responsible for the attenuation and/or loss of the ecGCx via ROS-activated matrix metalloproteinases (red-dashed arrows) and in turn, may result in the loss of the ecGCx that may contribute to an increase in aMt (black-dashed arrows). Moreover, mtROS (superoxide or hydrogen peroxide (H2O2)) could oxidize the essential fully reduced and essential tetrahydrobiopterin (BH4) cofactor to oxidized biopterin (BH3 and BH2) that will not enable eNOS to synthesize nitric oxide (NO) and results in eNOS uncoupling. eNOS uncoupling results in decreased bioavailable NO, as occurs in BEC*act/dys*. Importantly, the depicted bidirectional interaction could result in a vicious cycle. This vicious cycle results in blood–brain barrier (BBB) disruption with increased neurovascular unit (NVU) BEC permeability that would support the entry of excess fluid into the PVS, which could result in an EPVS. This vicious cycle could be interrupted by either preventing the accumulation of aMt (improved mitophagy) or preventing the dysfunction, attenuation, and/or loss (shedding) of the ecGCx. *BH3 and BH2* = *oxidized biopterin*; *NADPH Ox* = *nicotinamide adenine dinucleotide phosphate reduced oxidase*.

**Figure 14 medicina-59-00917-f014:**
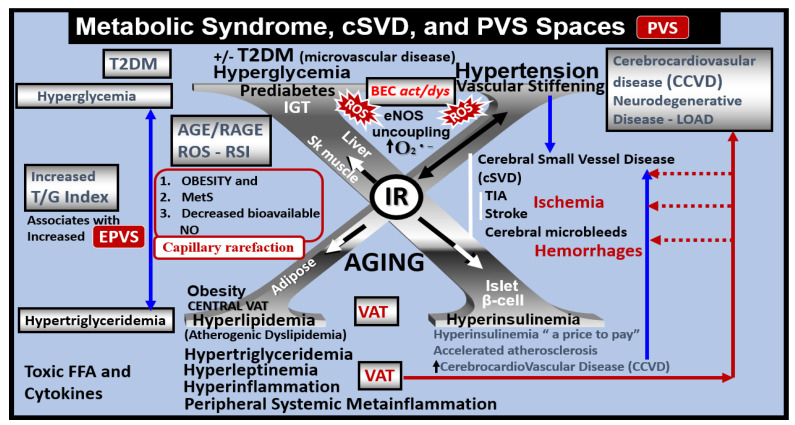
Metabolic syndrome (MetS), enlarged perivascular spaces (EPVS), and cerebral small vessel disease (SVD). The central X has four arms consisting of hyperlipidemia (lower left), hyperinsulinemia of insulin resistance (IR) (lower right), essential hypertension (upper right), and hyperglycemia (upper left). It is currently known that enlarged PVS (EPVS) are a biomarker of SVD. Visceral adipose tissue (VAT), increased triglyceride/glucose index (TG index), and hypertension are known to associate with SVD. Each of these four arms is either directly or indirectly associated with EPVS and SVD. Importantly, note that the triad of obesity, MetS, and decreased bioavailable nitric oxide (NO) are associated with capillary rarefaction and EPVS are known to be biomarkers of SVD. *AGE* = *advanced glycation end-products*; *RAGE* = *receptor for AGE*; *AGE/RAGE* = *advanced glycation end-products and its receptor interaction*; *BEC act/dys* = *brain endothelial cell activation and dysfunction*; *eNOS* = *endothelial nitric oxide synthase*; *FFA* = *free fatty acids–unsaturated long chain fatty acids*; *IGT* = *impaired glucose tolerance*; *LOAD* = *late-onset Alzheimer’s disease*; *O_2_ ^•−^* = *superoxide*; *ROS* = *reactive oxygen species*; *RSI* = *reactive species interactome*; *Sk* = *skeletal*; *T2DM* = *type 2 diabetes mellitus*; *TG Index* = *triglyceride/glucose index*; *TIA* = *transient ischemia attack*; *VAT* = *visceral adipose tissue*.

**Figure 15 medicina-59-00917-f015:**
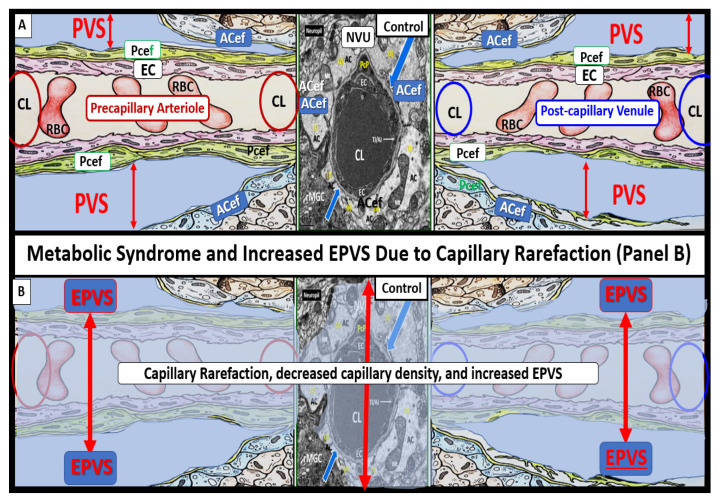
Obesity, metabolic syndrome (MetS), and enlarged perivascular spaces (EPVS) may be associated with brain capillary rarefaction. (**A**) demonstrates the normal precapillary arteriole (left), a true capillary (center), and a post-capillary venule (right) with their associated normal perivascular spaces (PVS). It is important to note that the pia matter lining abruptly stops at the true capillary and that the astrocyte end-feet directly abut the basement membrane (BM) of the neurovascular unit (NVU) mural cells (endothelial cell(s) (EC) and pericyte end-feet (Pcef)). (**B**) depicts the associated aberrant remodeling changes associated with the increased capillary rarefaction that is associated with obesity and MetS by utilizing a semi-transparent masking process. Note that as capillary rarefaction (capillary loss) is completed, there is a loss of the central capillary that runs through the PVS and how the PVS undergoes volume expansion due to the PVS now becoming only a fluid-filled space (light blue) similar to the PVS in (**A**), but now without a capillary running through the PVS. Additionally, note that capillary rarefaction also includes the NVU true capillary and how this also assists in the formation of EPVS when there is blood–brain barrier disruption. This depiction of how capillary rarefaction may be one of the multiple causes of EPVS fits nicely with the other causes that are thought to be important for the development of EPV, which include PVS obstruction, decreased pial artery, and arteriole pulsatility due to vascular stiffening and cerebral atrophy. Interestingly, the capillary rarefaction would contribute to regional ischemia and subsequent cerebral atrophy. *ACef* = *astrocyte end-feet*; *CL* = *capillary lumen*; *EC* = *endothelial cell*; *Pcef* = *pericyte end-feet*; *RBC* = *red blood cell*.

**Figure 16 medicina-59-00917-f016:**
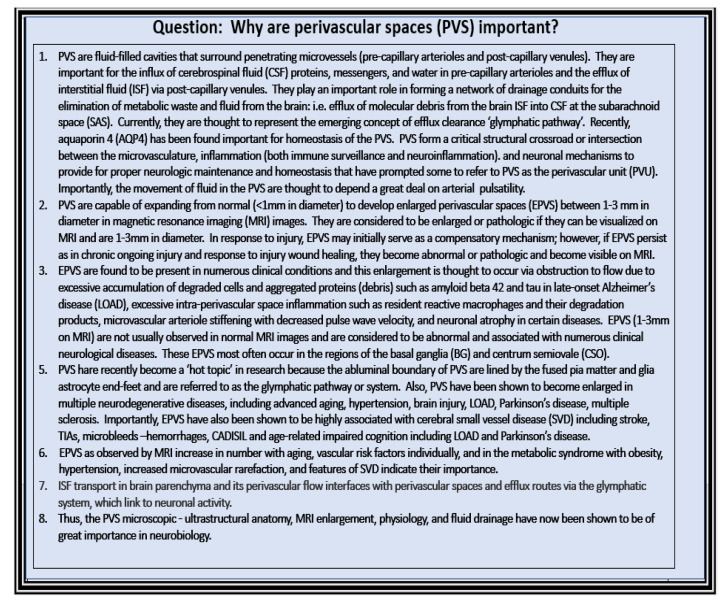
Eight core reasons why perivascular spaces are important.

## Data Availability

Data and materials will be provided upon reasonable request.

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
