# Peer review of "Why Are Perivascular Spaces Important?"

_medicina, 2023, doi:10.3390/medicina59050917_

Round 1

Reviewer 1 Report

Dear Authors,

I appreciate the Your valuable work, however, some corrections and clarifications should be introduced into the manuscript prior to publication.

Point 1. The “Reference” is not formatted according to the journal guidelines (https://www.mdpi.com/journal/medicina/instructions#references). I would recommend using “Zotero” software to correctly format all references in a semi-automatic way (use “Zotero” style file for “Multidisciplinary Digital Publishing Institute”; add sources by DOI – it will ease the whole process).

Point 2. The following format should be used for the “Author Contributions” section: "Conceptualization, X.X. and Y.Y.; Methodology, X.X.; Software, X.X.; Validation, X.X., Y.Y. and Z.Z.; Formal Analysis, X.X.; Investigation, X.X.; Resources, X.X.; Data Curation, X.X.; Writing – Original Draft Preparation, X.X.; Writing – Review & Editing, X.X.; Visualization, X.X.; Supervision, X.X.; Project Administration, X.X.; Funding Acquisition, Y.Y.”.

Point 3. Please, consider converting Figure 16 into text as it is extremely hard to read such a compressed block of text as well as there is no necessity for it to be a separate figure.

Point 4. All the Latin terms should be in italics.

Point 5. Please, consider providing more information about how results compare with previous studies in this area in order to contextualize findings within existing literature to broaden the bibliographic section.

Point 6. Remove the double spaces after in 37, 42, 50, 62, 66, 80, 82, 83, 86, 99, 103, 107, 110, 122, 125, 144, 158, 183, 198, 200, 201, 228, 234, 235, 237, 238, 241, 242, 255, 267, 290, 292, 295, 296, 303, 311, 324, 325, 357, 359, 377, 380, 385, 391,393, 395, 400, 408, 411, 416, 418, 420, 423, 426, 428, 431, 432, 442, 443, 466, 493, 515.

Faithfully Yours,

The Reviewer

Author Response

REVIEWER # 1:

I appreciate the Your valuable work, however, some corrections and clarifications should be introduced into the manuscript prior to publication.

Point 1. The “Reference” is not formatted according to the journal guidelines (https://www.mdpi.com/journal/medicina/instructions#references). I would recommend using “Zotero” software to correctly format all references in a semi-automatic way (use “Zotero” style file for “Multidisciplinary Digital Publishing Institute”; add sources by DOI – it will ease the whole process).  Authors wish to thank reviewer number 1 for these kind suggestions and recommendations.  However, we just heard from Ms. Snezana Vidakovic (Section Managing Editor) that the references will be reformatted according to the journal’s reference requirements and that editors will revise them in the final stage of processing the paper.  

Point 2. The following format should be used for the “Author Contributions” section: "Conceptualization, X.X. and Y.Y.; Methodology, X.X.; Software, X.X.; Validation, X.X., Y.Y. and Z.Z.; Formal Analysis, X.X.; Investigation, X.X.; Resources, X.X.; Data Curation, X.X.; Writing – Original Draft Preparation, X.X.; Writing – Review & Editing, X.X.; Visualization, X.X.; Supervision, X.X.; Project Administration, X.X.; Funding Acquisition, Y.Y.”.  These changes have been added to the manuscript see lines [529-535].

Point 3. Please, consider converting Figure 16 into text as it is extremely hard to read such a compressed block of text as well as there is no necessity for it to be a separate figure.  Authors would like to retain Figure 16. Authors agree with reviewer number 1 that this figure is compacted/compressed; however, its content is readily visualized and very easy to read when enlarged on the + tool icon in PDF and on the sliding bar in Word documents.  We feel the content in Figure 16 will assist in ppt presentations and teaching on large viewing screens, and assist many in detailing and teaching the 8 core reasons as to “Why PVS are Important”

Point 4. All the Latin terms should be in italics.  Upon review of the manuscript, we did not find any Latin terms that were used.  We utilized italics for act/dys following BECact/dys since we had used this terminology in previous publications as in reference number 43

Point 5. Please, consider providing more information about how results compare with previous studies in this area in order to contextualize findings within existing literature to broaden the bibliographic section.  This narrative review parallels many of the referenced published papers; however, authors have also utilized many TEM images in regard to PVS and EPVS, which are important to allow for a better understanding in regard to the associated ultrastructure remodeling.  Additionally, authors have provided many schematic illustrations to aid in the understanding of why PVS and EPVS are important.  See lines 480 to 494.

Point 6. Remove the double spaces after in 37, 42, 50, 62, 66, 80, 82, 83, 86, 99, 103, 107, 110, 122, 125, 144, 158, 183, 198, 200, 201, 228, 234, 235, 237, 238, 241, 242, 255, 267, 290, 292, 295, 296, 303, 311, 324, 325, 357, 359, 377, 380, 385, 391,393, 395, 400, 408, 411, 416, 418, 420, 423, 426, 428, 431, 432, 442, 443, 466, 493, 515.  These double spaces have now been corrected in the above multiple lines

Faithfully Yours,

The Reviewer

REVIEWER # 1:

I appreciate the Your valuable work, however, some corrections and clarifications should be introduced into the manuscript prior to publication.

Point 1. The “Reference” is not formatted according to the journal guidelines (https://www.mdpi.com/journal/medicina/instructions#references). I would recommend using “Zotero” software to correctly format all references in a semi-automatic way (use “Zotero” style file for “Multidisciplinary Digital Publishing Institute”; add sources by DOI – it will ease the whole process).  Authors wish to thank reviewer number 1 for these kind suggestions and recommendations.  However, we just heard from Ms. Snezana Vidakovic (Section Managing Editor) that the references will be reformatted according to the journal’s reference requirements and that editors will revise them in the final stage of processing the paper.  

Point 2. The following format should be used for the “Author Contributions” section: "Conceptualization, X.X. and Y.Y.; Methodology, X.X.; Software, X.X.; Validation, X.X., Y.Y. and Z.Z.; Formal Analysis, X.X.; Investigation, X.X.; Resources, X.X.; Data Curation, X.X.; Writing – Original Draft Preparation, X.X.; Writing – Review & Editing, X.X.; Visualization, X.X.; Supervision, X.X.; Project Administration, X.X.; Funding Acquisition, Y.Y.”.  These changes have been added to the manuscript see lines [529-535].

Point 3. Please, consider converting Figure 16 into text as it is extremely hard to read such a compressed block of text as well as there is no necessity for it to be a separate figure.  Authors would like to retain Figure 16. Authors agree with reviewer number 1 that this figure is compacted/compressed; however, its content is readily visualized and very easy to read when enlarged on the + tool icon in PDF and on the sliding bar in Word documents.  We feel the content in Figure 16 will assist in ppt presentations and teaching on large viewing screens, and assist many in detailing and teaching the 8 core reasons as to “Why PVS are Important”

Point 4. All the Latin terms should be in italics.  Upon review of the manuscript, we did not find any Latin terms that were used.  We utilized italics for act/dys following BECact/dys since we had used this terminology in previous publications as in reference number 43

Point 5. Please, consider providing more information about how results compare with previous studies in this area in order to contextualize findings within existing literature to broaden the bibliographic section.  This narrative review parallels many of the referenced published papers; however, authors have also utilized many TEM images in regard to PVS and EPVS, which are important to allow for a better understanding in regard to the associated ultrastructure remodeling.  Additionally, authors have provided many schematic illustrations to aid in the understanding of why PVS and EPVS are important.  See lines 480 to 494.

Point 6. Remove the double spaces after in 37, 42, 50, 62, 66, 80, 82, 83, 86, 99, 103, 107, 110, 122, 125, 144, 158, 183, 198, 200, 201, 228, 234, 235, 237, 238, 241, 242, 255, 267, 290, 292, 295, 296, 303, 311, 324, 325, 357, 359, 377, 380, 385, 391,393, 395, 400, 408, 411, 416, 418, 420, 423, 426, 428, 431, 432, 442, 443, 466, 493, 515.  These double spaces have now been corrected in the above multiple lines

Faithfully Yours,

The Reviewer

REVIEWER # 1:

I appreciate the Your valuable work, however, some corrections and clarifications should be introduced into the manuscript prior to publication.

Point 1. The “Reference” is not formatted according to the journal guidelines (https://www.mdpi.com/journal/medicina/instructions#references). I would recommend using “Zotero” software to correctly format all references in a semi-automatic way (use “Zotero” style file for “Multidisciplinary Digital Publishing Institute”; add sources by DOI – it will ease the whole process).  Authors wish to thank reviewer number 1 for these kind suggestions and recommendations.  However, we just heard from Ms. Snezana Vidakovic (Section Managing Editor) that the references will be reformatted according to the journal’s reference requirements and that editors will revise them in the final stage of processing the paper.  

Point 2. The following format should be used for the “Author Contributions” section: "Conceptualization, X.X. and Y.Y.; Methodology, X.X.; Software, X.X.; Validation, X.X., Y.Y. and Z.Z.; Formal Analysis, X.X.; Investigation, X.X.; Resources, X.X.; Data Curation, X.X.; Writing – Original Draft Preparation, X.X.; Writing – Review & Editing, X.X.; Visualization, X.X.; Supervision, X.X.; Project Administration, X.X.; Funding Acquisition, Y.Y.”.  These changes have been added to the manuscript see lines [529-535].

Point 3. Please, consider converting Figure 16 into text as it is extremely hard to read such a compressed block of text as well as there is no necessity for it to be a separate figure.  Authors would like to retain Figure 16. Authors agree with reviewer number 1 that this figure is compacted/compressed; however, its content is readily visualized and very easy to read when enlarged on the + tool icon in PDF and on the sliding bar in Word documents.  We feel the content in Figure 16 will assist in ppt presentations and teaching on large viewing screens, and assist many in detailing and teaching the 8 core reasons as to “Why PVS are Important”

Point 4. All the Latin terms should be in italics.  Upon review of the manuscript, we did not find any Latin terms that were used.  We utilized italics for act/dys following BECact/dys since we had used this terminology in previous publications as in reference number 43

Point 5. Please, consider providing more information about how results compare with previous studies in this area in order to contextualize findings within existing literature to broaden the bibliographic section.  This narrative review parallels many of the referenced published papers; however, authors have also utilized many TEM images in regard to PVS and EPVS, which are important to allow for a better understanding in regard to the associated ultrastructure remodeling.  Additionally, authors have provided many schematic illustrations to aid in the understanding of why PVS and EPVS are important.  See lines 480 to 494.

Point 6. Remove the double spaces after in 37, 42, 50, 62, 66, 80, 82, 83, 86, 99, 103, 107, 110, 122, 125, 144, 158, 183, 198, 200, 201, 228, 234, 235, 237, 238, 241, 242, 255, 267, 290, 292, 295, 296, 303, 311, 324, 325, 357, 359, 377, 380, 385, 391,393, 395, 400, 408, 411, 416, 418, 420, 423, 426, 428, 431, 432, 442, 443, 466, 493, 515.  These double spaces have now been corrected in the above multiple lines

Faithfully Yours,

The Reviewer

REVIEWER # 1:

I appreciate the Your valuable work, however, some corrections and clarifications should be introduced into the manuscript prior to publication.

Point 1. The “Reference” is not formatted according to the journal guidelines (https://www.mdpi.com/journal/medicina/instructions#references). I would recommend using “Zotero” software to correctly format all references in a semi-automatic way (use “Zotero” style file for “Multidisciplinary Digital Publishing Institute”; add sources by DOI – it will ease the whole process).  Authors wish to thank reviewer number 1 for these kind suggestions and recommendations.  However, we just heard from Ms. Snezana Vidakovic (Section Managing Editor) that the references will be reformatted according to the journal’s reference requirements and that editors will revise them in the final stage of processing the paper.  

Point 2. The following format should be used for the “Author Contributions” section: "Conceptualization, X.X. and Y.Y.; Methodology, X.X.; Software, X.X.; Validation, X.X., Y.Y. and Z.Z.; Formal Analysis, X.X.; Investigation, X.X.; Resources, X.X.; Data Curation, X.X.; Writing – Original Draft Preparation, X.X.; Writing – Review & Editing, X.X.; Visualization, X.X.; Supervision, X.X.; Project Administration, X.X.; Funding Acquisition, Y.Y.”.  These changes have been added to the manuscript see lines [529-535].

Point 3. Please, consider converting Figure 16 into text as it is extremely hard to read such a compressed block of text as well as there is no necessity for it to be a separate figure.  Authors would like to retain Figure 16. Authors agree with reviewer number 1 that this figure is compacted/compressed; however, its content is readily visualized and very easy to read when enlarged on the + tool icon in PDF and on the sliding bar in Word documents.  We feel the content in Figure 16 will assist in ppt presentations and teaching on large viewing screens, and assist many in detailing and teaching the 8 core reasons as to “Why PVS are Important”

Point 4. All the Latin terms should be in italics.  Upon review of the manuscript, we did not find any Latin terms that were used.  We utilized italics for act/dys following BECact/dys since we had used this terminology in previous publications as in reference number 43

Point 5. Please, consider providing more information about how results compare with previous studies in this area in order to contextualize findings within existing literature to broaden the bibliographic section.  This narrative review parallels many of the referenced published papers; however, authors have also utilized many TEM images in regard to PVS and EPVS, which are important to allow for a better understanding in regard to the associated ultrastructure remodeling.  Additionally, authors have provided many schematic illustrations to aid in the understanding of why PVS and EPVS are important.  See lines 480 to 494.

Point 6. Remove the double spaces after in 37, 42, 50, 62, 66, 80, 82, 83, 86, 99, 103, 107, 110, 122, 125, 144, 158, 183, 198, 200, 201, 228, 234, 235, 237, 238, 241, 242, 255, 267, 290, 292, 295, 296, 303, 311, 324, 325, 357, 359, 377, 380, 385, 391,393, 395, 400, 408, 411, 416, 418, 420, 423, 426, 428, 431, 432, 442, 443, 466, 493, 515.  These double spaces have now been corrected in the above multiple lines

Faithfully Yours,

The Reviewer

Reviewer 2 Report

The aim of this manuscript is to discuss the EPVS association with basal ganglia (BG), centrum semiovale (CSO), lacunes, WMH, and SVD; brain endothelial cell activation and dysfunction (BECact/dys) and BEC glycocalyx (ecGCx) shedding associated with EPVS, and SVD in section 3.; the metabolic syndrome (MetS), SVD, and PVS.

This manuscript shows rich content, providing a deep insight for some works: the study is within the journal’s scope, and I found it to be well-written, providing sufficient information. Even if the manuscript provides an organic overview, with a densely organized structure and based on well-synthetized evidence, there are some suggestions necessary to make the article complete and fully readable. For these reasons, the manuscript requires major changes.

Please find below an enumerated list of comments on my review of the manuscript:

INTRODUCTION:

The authors should provide a list of the abbreviations, used in the manuscript, before the introductive section.

LINE 42: It seem that the verb “are” does not agree with the subject. Consider changing the verb form.

LINE 92: Furthermore, recent studies reported that EPVS is also associated to a genetic predisposition to Alzheimer’s Disease (AD), providing evidence that the biological pathways affecting AD may also influence EPVS (see, for reference: Ciampa, I.; Operto, G.; Falcon, C.; Minguillon, C.; Castro de Moura, M.; Piñeyro, D.; Esteller, M.; Molinuevo, J.L.; Guigó, R.; Navarro, A.; Gispert, J.D.; Vilor-Tejedor, N.; for the ALFA Study. Genetic Predisposition to Alzheimer’s Disease Is Associated with Enlargement of Perivascular Spaces in Centrum Semiovale Region. Genes 202112, 825. https://doi.org/10.3390/genes12060825).

LINE 95: In this perspective, microscopy plays a crucial role, being a fundamental tool in clinical setting. If light microscopy (LM) is an essential tool for describing the most significant morphological changes of tissues, transmission electron microscopy (TEM) is a powerful instrument which provides ultrastructural evidence of tissues, to understand the physiological and pathological dynamics of a tissue (see, for reference: Torge, D.; Bernardi, S.; Arcangeli, M.; Bianchi, S. Histopathological Features of SARS-CoV-2 in Extrapulmonary Organ Infection: A Systematic Review of Literature. Pathogens 202211, 867. https://doi.org/10.3390/pathogens11080867). This is the major concern of this manuscript: the authors should discuss the importance of microscopy in clinical setting.

LINE 120: Microbleeds, intracerebral hemorrhages…

LINE 146: Initially, the response to injury wound healing mechanisms is protective. It seems that the verb “are” does not agree with the subject. Consider changing the verb form.

The main topic is interesting, and certainly of great clinical impact. As regards the originality and strengths of this manuscript, this is a significant contribute to the ongoing research on this topic, as it extends the research field on the EPVS association with basal ganglia (BG), centrum semiovale (CSO), lacunes, WMH, and SVD; brain endothelial cell activation and dysfunction (BECact/dys) and BEC glycocalyx (ecGCx) shedding associated with EPVS, and SVD in section 3.; the metabolic syndrome (MetS), SVD, and PVS. Overall, the contents are rich, and the authors also give their deep insight for some works.

There is a specific and detailed explanation for the evidence mentioned in this study: this is particularly significant, since the manuscript relies on a multitude of studies, to derive its conclusions.

The conclusion of this manuscript is perfectly in line with the main purpose of the paper: the authors have designed and conducted the study properly. As regards the conclusions, they are well written and present an adequate balance between the description of previous findings and the results presented by the authors.

In conclusion, this manuscript is densely presented and well organized, based on well-synthetized evidence. The authors were lucid in their style of writing, making it easy to read and understand the message, portrayed in the manuscript. Besides, the methodology design was appropriately implemented within the study. However, many of the topics are very concisely covered. This manuscript provided a comprehensive analysis of current knowledge in this field. Moreover, this research has futuristic importance and could be potential for future research. However, major concerns of this manuscript are with the introductive section: for these reasons, I have major comments for this section, for improvement before acceptance for publication. The article is accurate and provides relevant information on the topic and I have some major points to make, that may help to improve the quality of the current manuscript and maximize its scientific impact. I would accept this manuscript if the comments are addressed properly.

Minor editing of English Language is required.

Author Response

Reviewer # 2

The aim of this manuscript is to discuss the EPVS association with basal ganglia (BG), centrum semiovale (CSO), lacunes, WMH, and SVD; brain endothelial cell activation and dysfunction (BECact/dys) and BEC glycocalyx (ecGCx) shedding associated with EPVS, and SVD in section 3.; the metabolic syndrome (MetS), SVD, and PVS.

This manuscript shows rich content, providing a deep insight for some works: the study is within the journal’s scope, and I found it to be well-written, providing sufficient information. Even if the manuscript provides an organic overview, with a densely organized structure and based on well-synthetized evidence, there are some suggestions necessary to make the article complete and fully readable. For these reasons, the manuscript requires major changes.

Please find below an enumerated list of comments on my review of the manuscript:

INTRODUCTION:

The authors should provide a list of the abbreviations, used in the manuscript, before the introductive section. Authors wish to thank reviewer number 2 for these suggestions and we agree that adding abbreviations should be done. However, we just heard from Ms Snezana Vidakovic (Section Managing Editor) that we were to place the list of abbreviations after the conclusion per her directions so we have done that.  Please see lines [511-526]

LINE 42: It seem that the verb “are” does not agree with the subject. Please note that authors have now changed the opening sentence in the Introduction section to the following:  Perivascular spaces (PVS) pleural so that “are” is now the correct verb usage for this and other sentences to follow.

LINE 92: Furthermore, recent studies reported that EPVS are also associated to a genetic predisposition to Alzheimer’s Disease (AD), providing evidence that the biological pathways affecting AD may also influence EPVS (see, for reference: Ciampa, I.; Operto, G.; Falcon, C.; Minguillon, C.; Castro de Moura, M.; Piñeyro, D.; Esteller, M.; Molinuevo, J.L.; Guigó, R.; Navarro, A.; Gispert, J.D.; Vilor-Tejedor, N.; for the ALFA Study. Genetic Predisposition to Alzheimer’s Disease Is Associated with Enlargement of Perivascular Spaces in Centrum Semiovale Region. Genes 202112, 825. https://doi.org/10.3390/genes12060825).  Please note that this comment and new reference were added as reference 7 with the references changed accordingly in lines [113-115]

LINE 95: In this perspective, microscopy plays a crucial role, being a fundamental tool in clinical setting. If light microscopy (LM) is an essential tool for describing the most significant morphological changes of tissues, transmission electron microscopy (TEM) is a powerful instrument which provides ultrastructural evidence of tissues, to understand the physiological and pathological dynamics of a tissue (see, for reference: Torge, D.; Bernardi, S.; Arcangeli, M.; Bianchi, S. Histopathological Features of SARS-CoV-2 in Extrapulmonary Organ Infection: A Systematic Review of Literature. Pathogens 202211, 867. https://doi.org/10.3390/pathogens11080867). This is the major concern of this manuscript: the authors should discuss the importance of microscopy in clinical setting.  Authors are extremely grateful for both of these above recommendations and wish to thank reviewer number 2 as we feel this will further support the use of TEM and microscopy in general, to better understand PVS and their enlargement.  Please note the following insertion: “In this perspective, microscopy plays a crucial role, being a fundamental tool in the clinical setting. Light microscopy is an essential tool for describing the significant morphological changes of tissues and TEM is a powerful instrument, which provides ultrastructural evidence of tissue remodeling, to understand the physiological and pathological dynamics of PVS development and remodeling” in lines 115-119.

LINE 120: Microbleeds, intracerebral hemorrhages… Authors have now added intracerebral hemorrhages…

LINE 146: Initially, the response to injury wound healing mechanisms is protective. It seems that the verb “are” does not agree with the subject.  This has now been changed  to … wound healing mechanisms is

The main topic is interesting, and certainly of great clinical impact. As regards the originality and strengths of this manuscript, this is a significant contribute to the ongoing research on this topic, as it extends the research field on the EPVS association with basal ganglia (BG), centrum semiovale (CSO), lacunes, WMH, and SVD; brain endothelial cell activation and dysfunction (BECact/dys) and BEC glycocalyx (ecGCx) shedding associated with EPVS, and SVD in section 3.; the metabolic syndrome (MetS), SVD, and PVS. Overall, the contents are rich, and the authors also give their deep insight for some works.

There is a specific and detailed explanation for the evidence mentioned in this study: this is particularly significant, since the manuscript relies on a multitude of studies, to derive its conclusions.

The conclusion of this manuscript is perfectly in line with the main purpose of the paper: the authors have designed and conducted the study properly. As regards the conclusions, they are well written and present an adequate balance between the description of previous findings and the results presented by the authors.

In conclusion, this manuscript is densely presented and well organized, based on well-synthetized evidence. The authors were lucid in their style of writing, making it easy to read and understand the message, portrayed in the manuscript. Besides, the methodology design was appropriately implemented within the study. However, many of the topics are very concisely covered. This manuscript provided a comprehensive analysis of current knowledge in this field. Moreover, this research has futuristic importance and could be potential for future research. However, major concerns of this manuscript are with the introductive section: for these reasons, I have major comments for this section, for improvement before acceptance for publication. The article is accurate and provides relevant information on the topic and I have some major points to make, that may help to improve the quality of the current manuscript and maximize its scientific impact. I would accept this manuscript if the comments are addressed properly.  The authors wish to kindly thank reviewer number 2 for these kind comments and appropriate recommendations to improve the manuscript.  Thank you so very much for your precious time and knowledge for making this manuscript better for our readers.  We have made the changes that you have recommended  see lined manuscript with blue lettering and also reference changes in blue lettering.

Reviewer # 2

The aim of this manuscript is to discuss the EPVS association with basal ganglia (BG), centrum semiovale (CSO), lacunes, WMH, and SVD; brain endothelial cell activation and dysfunction (BECact/dys) and BEC glycocalyx (ecGCx) shedding associated with EPVS, and SVD in section 3.; the metabolic syndrome (MetS), SVD, and PVS.

This manuscript shows rich content, providing a deep insight for some works: the study is within the journal’s scope, and I found it to be well-written, providing sufficient information. Even if the manuscript provides an organic overview, with a densely organized structure and based on well-synthetized evidence, there are some suggestions necessary to make the article complete and fully readable. For these reasons, the manuscript requires major changes.

Please find below an enumerated list of comments on my review of the manuscript:

INTRODUCTION:

The authors should provide a list of the abbreviations, used in the manuscript, before the introductive section. Authors wish to thank reviewer number 2 for these suggestions and we agree that adding abbreviations should be done. However, we just heard from Ms Snezana Vidakovic (Section Managing Editor) that we were to place the list of abbreviations after the conclusion per her directions so we have done that.  Please see lines [511-526]

LINE 42: It seem that the verb “are” does not agree with the subject. Please note that authors have now changed the opening sentence in the Introduction section to the following:  Perivascular spaces (PVS) pleural so that “are” is now the correct verb usage for this and other sentences to follow.

LINE 92: Furthermore, recent studies reported that EPVS are also associated to a genetic predisposition to Alzheimer’s Disease (AD), providing evidence that the biological pathways affecting AD may also influence EPVS (see, for reference: Ciampa, I.; Operto, G.; Falcon, C.; Minguillon, C.; Castro de Moura, M.; Piñeyro, D.; Esteller, M.; Molinuevo, J.L.; Guigó, R.; Navarro, A.; Gispert, J.D.; Vilor-Tejedor, N.; for the ALFA Study. Genetic Predisposition to Alzheimer’s Disease Is Associated with Enlargement of Perivascular Spaces in Centrum Semiovale Region. Genes 202112, 825. https://doi.org/10.3390/genes12060825).  Please note that this comment and new reference were added as reference 7 with the references changed accordingly in lines [113-115]

LINE 95: In this perspective, microscopy plays a crucial role, being a fundamental tool in clinical setting. If light microscopy (LM) is an essential tool for describing the most significant morphological changes of tissues, transmission electron microscopy (TEM) is a powerful instrument which provides ultrastructural evidence of tissues, to understand the physiological and pathological dynamics of a tissue (see, for reference: Torge, D.; Bernardi, S.; Arcangeli, M.; Bianchi, S. Histopathological Features of SARS-CoV-2 in Extrapulmonary Organ Infection: A Systematic Review of Literature. Pathogens 202211, 867. https://doi.org/10.3390/pathogens11080867). This is the major concern of this manuscript: the authors should discuss the importance of microscopy in clinical setting.  Authors are extremely grateful for both of these above recommendations and wish to thank reviewer number 2 as we feel this will further support the use of TEM and microscopy in general, to better understand PVS and their enlargement.  Please note the following insertion: “In this perspective, microscopy plays a crucial role, being a fundamental tool in the clinical setting. Light microscopy is an essential tool for describing the significant morphological changes of tissues and TEM is a powerful instrument, which provides ultrastructural evidence of tissue remodeling, to understand the physiological and pathological dynamics of PVS development and remodeling” in lines 115-119.

LINE 120: Microbleeds, intracerebral hemorrhages… Authors have now added intracerebral hemorrhages…

LINE 146: Initially, the response to injury wound healing mechanisms is protective. It seems that the verb “are” does not agree with the subject.  This has now been changed  to … wound healing mechanisms is

The main topic is interesting, and certainly of great clinical impact. As regards the originality and strengths of this manuscript, this is a significant contribute to the ongoing research on this topic, as it extends the research field on the EPVS association with basal ganglia (BG), centrum semiovale (CSO), lacunes, WMH, and SVD; brain endothelial cell activation and dysfunction (BECact/dys) and BEC glycocalyx (ecGCx) shedding associated with EPVS, and SVD in section 3.; the metabolic syndrome (MetS), SVD, and PVS. Overall, the contents are rich, and the authors also give their deep insight for some works.

There is a specific and detailed explanation for the evidence mentioned in this study: this is particularly significant, since the manuscript relies on a multitude of studies, to derive its conclusions.

The conclusion of this manuscript is perfectly in line with the main purpose of the paper: the authors have designed and conducted the study properly. As regards the conclusions, they are well written and present an adequate balance between the description of previous findings and the results presented by the authors.

In conclusion, this manuscript is densely presented and well organized, based on well-synthetized evidence. The authors were lucid in their style of writing, making it easy to read and understand the message, portrayed in the manuscript. Besides, the methodology design was appropriately implemented within the study. However, many of the topics are very concisely covered. This manuscript provided a comprehensive analysis of current knowledge in this field. Moreover, this research has futuristic importance and could be potential for future research. However, major concerns of this manuscript are with the introductive section: for these reasons, I have major comments for this section, for improvement before acceptance for publication. The article is accurate and provides relevant information on the topic and I have some major points to make, that may help to improve the quality of the current manuscript and maximize its scientific impact. I would accept this manuscript if the comments are addressed properly.  The authors wish to kindly thank reviewer number 2 for these kind comments and appropriate recommendations to improve the manuscript.  Thank you so very much for your precious time and knowledge for making this manuscript better for our readers.  We have made the changes that you have recommended  see lined manuscript with blue lettering and also reference changes in blue lettering.

Reviewer # 2

The aim of this manuscript is to discuss the EPVS association with basal ganglia (BG), centrum semiovale (CSO), lacunes, WMH, and SVD; brain endothelial cell activation and dysfunction (BECact/dys) and BEC glycocalyx (ecGCx) shedding associated with EPVS, and SVD in section 3.; the metabolic syndrome (MetS), SVD, and PVS.

This manuscript shows rich content, providing a deep insight for some works: the study is within the journal’s scope, and I found it to be well-written, providing sufficient information. Even if the manuscript provides an organic overview, with a densely organized structure and based on well-synthetized evidence, there are some suggestions necessary to make the article complete and fully readable. For these reasons, the manuscript requires major changes.

Please find below an enumerated list of comments on my review of the manuscript:

INTRODUCTION:

The authors should provide a list of the abbreviations, used in the manuscript, before the introductive section. Authors wish to thank reviewer number 2 for these suggestions and we agree that adding abbreviations should be done. However, we just heard from Ms Snezana Vidakovic (Section Managing Editor) that we were to place the list of abbreviations after the conclusion per her directions so we have done that.  Please see lines [511-526]

LINE 42: It seem that the verb “are” does not agree with the subject. Please note that authors have now changed the opening sentence in the Introduction section to the following:  Perivascular spaces (PVS) pleural so that “are” is now the correct verb usage for this and other sentences to follow.

LINE 92: Furthermore, recent studies reported that EPVS are also associated to a genetic predisposition to Alzheimer’s Disease (AD), providing evidence that the biological pathways affecting AD may also influence EPVS (see, for reference: Ciampa, I.; Operto, G.; Falcon, C.; Minguillon, C.; Castro de Moura, M.; Piñeyro, D.; Esteller, M.; Molinuevo, J.L.; Guigó, R.; Navarro, A.; Gispert, J.D.; Vilor-Tejedor, N.; for the ALFA Study. Genetic Predisposition to Alzheimer’s Disease Is Associated with Enlargement of Perivascular Spaces in Centrum Semiovale Region. Genes 202112, 825. https://doi.org/10.3390/genes12060825).  Please note that this comment and new reference were added as reference 7 with the references changed accordingly in lines [113-115]

LINE 95: In this perspective, microscopy plays a crucial role, being a fundamental tool in clinical setting. If light microscopy (LM) is an essential tool for describing the most significant morphological changes of tissues, transmission electron microscopy (TEM) is a powerful instrument which provides ultrastructural evidence of tissues, to understand the physiological and pathological dynamics of a tissue (see, for reference: Torge, D.; Bernardi, S.; Arcangeli, M.; Bianchi, S. Histopathological Features of SARS-CoV-2 in Extrapulmonary Organ Infection: A Systematic Review of Literature. Pathogens 202211, 867. https://doi.org/10.3390/pathogens11080867). This is the major concern of this manuscript: the authors should discuss the importance of microscopy in clinical setting.  Authors are extremely grateful for both of these above recommendations and wish to thank reviewer number 2 as we feel this will further support the use of TEM and microscopy in general, to better understand PVS and their enlargement.  Please note the following insertion: “In this perspective, microscopy plays a crucial role, being a fundamental tool in the clinical setting. Light microscopy is an essential tool for describing the significant morphological changes of tissues and TEM is a powerful instrument, which provides ultrastructural evidence of tissue remodeling, to understand the physiological and pathological dynamics of PVS development and remodeling” in lines 115-119.

LINE 120: Microbleeds, intracerebral hemorrhages… Authors have now added intracerebral hemorrhages…

LINE 146: Initially, the response to injury wound healing mechanisms is protective. It seems that the verb “are” does not agree with the subject.  This has now been changed  to … wound healing mechanisms is

The main topic is interesting, and certainly of great clinical impact. As regards the originality and strengths of this manuscript, this is a significant contribute to the ongoing research on this topic, as it extends the research field on the EPVS association with basal ganglia (BG), centrum semiovale (CSO), lacunes, WMH, and SVD; brain endothelial cell activation and dysfunction (BECact/dys) and BEC glycocalyx (ecGCx) shedding associated with EPVS, and SVD in section 3.; the metabolic syndrome (MetS), SVD, and PVS. Overall, the contents are rich, and the authors also give their deep insight for some works.

There is a specific and detailed explanation for the evidence mentioned in this study: this is particularly significant, since the manuscript relies on a multitude of studies, to derive its conclusions.

The conclusion of this manuscript is perfectly in line with the main purpose of the paper: the authors have designed and conducted the study properly. As regards the conclusions, they are well written and present an adequate balance between the description of previous findings and the results presented by the authors.

In conclusion, this manuscript is densely presented and well organized, based on well-synthetized evidence. The authors were lucid in their style of writing, making it easy to read and understand the message, portrayed in the manuscript. Besides, the methodology design was appropriately implemented within the study. However, many of the topics are very concisely covered. This manuscript provided a comprehensive analysis of current knowledge in this field. Moreover, this research has futuristic importance and could be potential for future research. However, major concerns of this manuscript are with the introductive section: for these reasons, I have major comments for this section, for improvement before acceptance for publication. The article is accurate and provides relevant information on the topic and I have some major points to make, that may help to improve the quality of the current manuscript and maximize its scientific impact. I would accept this manuscript if the comments are addressed properly.  The authors wish to kindly thank reviewer number 2 for these kind comments and appropriate recommendations to improve the manuscript.  Thank you so very much for your precious time and knowledge for making this manuscript better for our readers.  We have made the changes that you have recommended  see lined manuscript with blue lettering and also reference changes in blue lettering.

Reviewer # 2

The aim of this manuscript is to discuss the EPVS association with basal ganglia (BG), centrum semiovale (CSO), lacunes, WMH, and SVD; brain endothelial cell activation and dysfunction (BECact/dys) and BEC glycocalyx (ecGCx) shedding associated with EPVS, and SVD in section 3.; the metabolic syndrome (MetS), SVD, and PVS.

This manuscript shows rich content, providing a deep insight for some works: the study is within the journal’s scope, and I found it to be well-written, providing sufficient information. Even if the manuscript provides an organic overview, with a densely organized structure and based on well-synthetized evidence, there are some suggestions necessary to make the article complete and fully readable. For these reasons, the manuscript requires major changes.

Please find below an enumerated list of comments on my review of the manuscript:

INTRODUCTION:

The authors should provide a list of the abbreviations, used in the manuscript, before the introductive section. Authors wish to thank reviewer number 2 for these suggestions and we agree that adding abbreviations should be done. However, we just heard from Ms Snezana Vidakovic (Section Managing Editor) that we were to place the list of abbreviations after the conclusion per her directions so we have done that.  Please see lines [511-526]

LINE 42: It seem that the verb “are” does not agree with the subject. Please note that authors have now changed the opening sentence in the Introduction section to the following:  Perivascular spaces (PVS) pleural so that “are” is now the correct verb usage for this and other sentences to follow.

LINE 92: Furthermore, recent studies reported that EPVS are also associated to a genetic predisposition to Alzheimer’s Disease (AD), providing evidence that the biological pathways affecting AD may also influence EPVS (see, for reference: Ciampa, I.; Operto, G.; Falcon, C.; Minguillon, C.; Castro de Moura, M.; Piñeyro, D.; Esteller, M.; Molinuevo, J.L.; Guigó, R.; Navarro, A.; Gispert, J.D.; Vilor-Tejedor, N.; for the ALFA Study. Genetic Predisposition to Alzheimer’s Disease Is Associated with Enlargement of Perivascular Spaces in Centrum Semiovale Region. Genes 202112, 825. https://doi.org/10.3390/genes12060825).  Please note that this comment and new reference were added as reference 7 with the references changed accordingly in lines [113-115]

LINE 95: In this perspective, microscopy plays a crucial role, being a fundamental tool in clinical setting. If light microscopy (LM) is an essential tool for describing the most significant morphological changes of tissues, transmission electron microscopy (TEM) is a powerful instrument which provides ultrastructural evidence of tissues, to understand the physiological and pathological dynamics of a tissue (see, for reference: Torge, D.; Bernardi, S.; Arcangeli, M.; Bianchi, S. Histopathological Features of SARS-CoV-2 in Extrapulmonary Organ Infection: A Systematic Review of Literature. Pathogens 202211, 867. https://doi.org/10.3390/pathogens11080867). This is the major concern of this manuscript: the authors should discuss the importance of microscopy in clinical setting.  Authors are extremely grateful for both of these above recommendations and wish to thank reviewer number 2 as we feel this will further support the use of TEM and microscopy in general, to better understand PVS and their enlargement.  Please note the following insertion: “In this perspective, microscopy plays a crucial role, being a fundamental tool in the clinical setting. Light microscopy is an essential tool for describing the significant morphological changes of tissues and TEM is a powerful instrument, which provides ultrastructural evidence of tissue remodeling, to understand the physiological and pathological dynamics of PVS development and remodeling” in lines 115-119.

LINE 120: Microbleeds, intracerebral hemorrhages… Authors have now added intracerebral hemorrhages…

LINE 146: Initially, the response to injury wound healing mechanisms is protective. It seems that the verb “are” does not agree with the subject.  This has now been changed  to … wound healing mechanisms is

The main topic is interesting, and certainly of great clinical impact. As regards the originality and strengths of this manuscript, this is a significant contribute to the ongoing research on this topic, as it extends the research field on the EPVS association with basal ganglia (BG), centrum semiovale (CSO), lacunes, WMH, and SVD; brain endothelial cell activation and dysfunction (BECact/dys) and BEC glycocalyx (ecGCx) shedding associated with EPVS, and SVD in section 3.; the metabolic syndrome (MetS), SVD, and PVS. Overall, the contents are rich, and the authors also give their deep insight for some works.

There is a specific and detailed explanation for the evidence mentioned in this study: this is particularly significant, since the manuscript relies on a multitude of studies, to derive its conclusions.

The conclusion of this manuscript is perfectly in line with the main purpose of the paper: the authors have designed and conducted the study properly. As regards the conclusions, they are well written and present an adequate balance between the description of previous findings and the results presented by the authors.

In conclusion, this manuscript is densely presented and well organized, based on well-synthetized evidence. The authors were lucid in their style of writing, making it easy to read and understand the message, portrayed in the manuscript. Besides, the methodology design was appropriately implemented within the study. However, many of the topics are very concisely covered. This manuscript provided a comprehensive analysis of current knowledge in this field. Moreover, this research has futuristic importance and could be potential for future research. However, major concerns of this manuscript are with the introductive section: for these reasons, I have major comments for this section, for improvement before acceptance for publication. The article is accurate and provides relevant information on the topic and I have some major points to make, that may help to improve the quality of the current manuscript and maximize its scientific impact. I would accept this manuscript if the comments are addressed properly.  The authors wish to kindly thank reviewer number 2 for these kind comments and appropriate recommendations to improve the manuscript.  Thank you so very much for your precious time and knowledge for making this manuscript better for our readers.  We have made the changes that you have recommended  see lined manuscript with blue lettering and also reference changes in blue lettering.

Reviewer # 2

The aim of this manuscript is to discuss the EPVS association with basal ganglia (BG), centrum semiovale (CSO), lacunes, WMH, and SVD; brain endothelial cell activation and dysfunction (BECact/dys) and BEC glycocalyx (ecGCx) shedding associated with EPVS, and SVD in section 3.; the metabolic syndrome (MetS), SVD, and PVS.

This manuscript shows rich content, providing a deep insight for some works: the study is within the journal’s scope, and I found it to be well-written, providing sufficient information. Even if the manuscript provides an organic overview, with a densely organized structure and based on well-synthetized evidence, there are some suggestions necessary to make the article complete and fully readable. For these reasons, the manuscript requires major changes.

Please find below an enumerated list of comments on my review of the manuscript:

INTRODUCTION:

The authors should provide a list of the abbreviations, used in the manuscript, before the introductive section. Authors wish to thank reviewer number 2 for these suggestions and we agree that adding abbreviations should be done. However, we just heard from Ms Snezana Vidakovic (Section Managing Editor) that we were to place the list of abbreviations after the conclusion per her directions so we have done that.  Please see lines [511-526]

LINE 42: It seem that the verb “are” does not agree with the subject. Please note that authors have now changed the opening sentence in the Introduction section to the following:  Perivascular spaces (PVS) pleural so that “are” is now the correct verb usage for this and other sentences to follow.

LINE 92: Furthermore, recent studies reported that EPVS are also associated to a genetic predisposition to Alzheimer’s Disease (AD), providing evidence that the biological pathways affecting AD may also influence EPVS (see, for reference: Ciampa, I.; Operto, G.; Falcon, C.; Minguillon, C.; Castro de Moura, M.; Piñeyro, D.; Esteller, M.; Molinuevo, J.L.; Guigó, R.; Navarro, A.; Gispert, J.D.; Vilor-Tejedor, N.; for the ALFA Study. Genetic Predisposition to Alzheimer’s Disease Is Associated with Enlargement of Perivascular Spaces in Centrum Semiovale Region. Genes 202112, 825. https://doi.org/10.3390/genes12060825).  Please note that this comment and new reference were added as reference 7 with the references changed accordingly in lines [113-115]

LINE 95: In this perspective, microscopy plays a crucial role, being a fundamental tool in clinical setting. If light microscopy (LM) is an essential tool for describing the most significant morphological changes of tissues, transmission electron microscopy (TEM) is a powerful instrument which provides ultrastructural evidence of tissues, to understand the physiological and pathological dynamics of a tissue (see, for reference: Torge, D.; Bernardi, S.; Arcangeli, M.; Bianchi, S. Histopathological Features of SARS-CoV-2 in Extrapulmonary Organ Infection: A Systematic Review of Literature. Pathogens 202211, 867. https://doi.org/10.3390/pathogens11080867). This is the major concern of this manuscript: the authors should discuss the importance of microscopy in clinical setting.  Authors are extremely grateful for both of these above recommendations and wish to thank reviewer number 2 as we feel this will further support the use of TEM and microscopy in general, to better understand PVS and their enlargement.  Please note the following insertion: “In this perspective, microscopy plays a crucial role, being a fundamental tool in the clinical setting. Light microscopy is an essential tool for describing the significant morphological changes of tissues and TEM is a powerful instrument, which provides ultrastructural evidence of tissue remodeling, to understand the physiological and pathological dynamics of PVS development and remodeling” in lines 115-119.

LINE 120: Microbleeds, intracerebral hemorrhages… Authors have now added intracerebral hemorrhages…

LINE 146: Initially, the response to injury wound healing mechanisms is protective. It seems that the verb “are” does not agree with the subject.  This has now been changed  to … wound healing mechanisms is

The main topic is interesting, and certainly of great clinical impact. As regards the originality and strengths of this manuscript, this is a significant contribute to the ongoing research on this topic, as it extends the research field on the EPVS association with basal ganglia (BG), centrum semiovale (CSO), lacunes, WMH, and SVD; brain endothelial cell activation and dysfunction (BECact/dys) and BEC glycocalyx (ecGCx) shedding associated with EPVS, and SVD in section 3.; the metabolic syndrome (MetS), SVD, and PVS. Overall, the contents are rich, and the authors also give their deep insight for some works.

There is a specific and detailed explanation for the evidence mentioned in this study: this is particularly significant, since the manuscript relies on a multitude of studies, to derive its conclusions.

The conclusion of this manuscript is perfectly in line with the main purpose of the paper: the authors have designed and conducted the study properly. As regards the conclusions, they are well written and present an adequate balance between the description of previous findings and the results presented by the authors.

In conclusion, this manuscript is densely presented and well organized, based on well-synthetized evidence. The authors were lucid in their style of writing, making it easy to read and understand the message, portrayed in the manuscript. Besides, the methodology design was appropriately implemented within the study. However, many of the topics are very concisely covered. This manuscript provided a comprehensive analysis of current knowledge in this field. Moreover, this research has futuristic importance and could be potential for future research. However, major concerns of this manuscript are with the introductive section: for these reasons, I have major comments for this section, for improvement before acceptance for publication. The article is accurate and provides relevant information on the topic and I have some major points to make, that may help to improve the quality of the current manuscript and maximize its scientific impact. I would accept this manuscript if the comments are addressed properly.  The authors wish to kindly thank reviewer number 2 for these kind comments and appropriate recommendations to improve the manuscript.  Thank you so very much for your precious time and knowledge for making this manuscript better for our readers.  We have made the changes that you have recommended  see lined manuscript with blue lettering and also reference changes in blue lettering.

Reviewer # 2

The aim of this manuscript is to discuss the EPVS association with basal ganglia (BG), centrum semiovale (CSO), lacunes, WMH, and SVD; brain endothelial cell activation and dysfunction (BECact/dys) and BEC glycocalyx (ecGCx) shedding associated with EPVS, and SVD in section 3.; the metabolic syndrome (MetS), SVD, and PVS.

This manuscript shows rich content, providing a deep insight for some works: the study is within the journal’s scope, and I found it to be well-written, providing sufficient information. Even if the manuscript provides an organic overview, with a densely organized structure and based on well-synthetized evidence, there are some suggestions necessary to make the article complete and fully readable. For these reasons, the manuscript requires major changes.

Please find below an enumerated list of comments on my review of the manuscript:

INTRODUCTION:

The authors should provide a list of the abbreviations, used in the manuscript, before the introductive section. Authors wish to thank reviewer number 2 for these suggestions and we agree that adding abbreviations should be done. However, we just heard from Ms Snezana Vidakovic (Section Managing Editor) that we were to place the list of abbreviations after the conclusion per her directions so we have done that.  Please see lines [511-526]

LINE 42: It seem that the verb “are” does not agree with the subject. Please note that authors have now changed the opening sentence in the Introduction section to the following:  Perivascular spaces (PVS) pleural so that “are” is now the correct verb usage for this and other sentences to follow.

LINE 92: Furthermore, recent studies reported that EPVS are also associated to a genetic predisposition to Alzheimer’s Disease (AD), providing evidence that the biological pathways affecting AD may also influence EPVS (see, for reference: Ciampa, I.; Operto, G.; Falcon, C.; Minguillon, C.; Castro de Moura, M.; Piñeyro, D.; Esteller, M.; Molinuevo, J.L.; Guigó, R.; Navarro, A.; Gispert, J.D.; Vilor-Tejedor, N.; for the ALFA Study. Genetic Predisposition to Alzheimer’s Disease Is Associated with Enlargement of Perivascular Spaces in Centrum Semiovale Region. Genes 202112, 825. https://doi.org/10.3390/genes12060825).  Please note that this comment and new reference were added as reference 7 with the references changed accordingly in lines [113-115]

LINE 95: In this perspective, microscopy plays a crucial role, being a fundamental tool in clinical setting. If light microscopy (LM) is an essential tool for describing the most significant morphological changes of tissues, transmission electron microscopy (TEM) is a powerful instrument which provides ultrastructural evidence of tissues, to understand the physiological and pathological dynamics of a tissue (see, for reference: Torge, D.; Bernardi, S.; Arcangeli, M.; Bianchi, S. Histopathological Features of SARS-CoV-2 in Extrapulmonary Organ Infection: A Systematic Review of Literature. Pathogens 202211, 867. https://doi.org/10.3390/pathogens11080867). This is the major concern of this manuscript: the authors should discuss the importance of microscopy in clinical setting.  Authors are extremely grateful for both of these above recommendations and wish to thank reviewer number 2 as we feel this will further support the use of TEM and microscopy in general, to better understand PVS and their enlargement.  Please note the following insertion: “In this perspective, microscopy plays a crucial role, being a fundamental tool in the clinical setting. Light microscopy is an essential tool for describing the significant morphological changes of tissues and TEM is a powerful instrument, which provides ultrastructural evidence of tissue remodeling, to understand the physiological and pathological dynamics of PVS development and remodeling” in lines 115-119.

LINE 120: Microbleeds, intracerebral hemorrhages… Authors have now added intracerebral hemorrhages…

LINE 146: Initially, the response to injury wound healing mechanisms is protective. It seems that the verb “are” does not agree with the subject.  This has now been changed  to … wound healing mechanisms is

The main topic is interesting, and certainly of great clinical impact. As regards the originality and strengths of this manuscript, this is a significant contribute to the ongoing research on this topic, as it extends the research field on the EPVS association with basal ganglia (BG), centrum semiovale (CSO), lacunes, WMH, and SVD; brain endothelial cell activation and dysfunction (BECact/dys) and BEC glycocalyx (ecGCx) shedding associated with EPVS, and SVD in section 3.; the metabolic syndrome (MetS), SVD, and PVS. Overall, the contents are rich, and the authors also give their deep insight for some works.

There is a specific and detailed explanation for the evidence mentioned in this study: this is particularly significant, since the manuscript relies on a multitude of studies, to derive its conclusions.

The conclusion of this manuscript is perfectly in line with the main purpose of the paper: the authors have designed and conducted the study properly. As regards the conclusions, they are well written and present an adequate balance between the description of previous findings and the results presented by the authors.

In conclusion, this manuscript is densely presented and well organized, based on well-synthetized evidence. The authors were lucid in their style of writing, making it easy to read and understand the message, portrayed in the manuscript. Besides, the methodology design was appropriately implemented within the study. However, many of the topics are very concisely covered. This manuscript provided a comprehensive analysis of current knowledge in this field. Moreover, this research has futuristic importance and could be potential for future research. However, major concerns of this manuscript are with the introductive section: for these reasons, I have major comments for this section, for improvement before acceptance for publication. The article is accurate and provides relevant information on the topic and I have some major points to make, that may help to improve the quality of the current manuscript and maximize its scientific impact. I would accept this manuscript if the comments are addressed properly.  The authors wish to kindly thank reviewer number 2 for these kind comments and appropriate recommendations to improve the manuscript.  Thank you so very much for your precious time and knowledge for making this manuscript better for our readers.  We have made the changes that you have recommended  see lined manuscript with blue lettering and also reference changes in blue lettering.

Reviewer # 2

The aim of this manuscript is to discuss the EPVS association with basal ganglia (BG), centrum semiovale (CSO), lacunes, WMH, and SVD; brain endothelial cell activation and dysfunction (BECact/dys) and BEC glycocalyx (ecGCx) shedding associated with EPVS, and SVD in section 3.; the metabolic syndrome (MetS), SVD, and PVS.

This manuscript shows rich content, providing a deep insight for some works: the study is within the journal’s scope, and I found it to be well-written, providing sufficient information. Even if the manuscript provides an organic overview, with a densely organized structure and based on well-synthetized evidence, there are some suggestions necessary to make the article complete and fully readable. For these reasons, the manuscript requires major changes.

Please find below an enumerated list of comments on my review of the manuscript:

INTRODUCTION:

The authors should provide a list of the abbreviations, used in the manuscript, before the introductive section. Authors wish to thank reviewer number 2 for these suggestions and we agree that adding abbreviations should be done. However, we just heard from Ms Snezana Vidakovic (Section Managing Editor) that we were to place the list of abbreviations after the conclusion per her directions so we have done that.  Please see lines [511-526]

LINE 42: It seem that the verb “are” does not agree with the subject. Please note that authors have now changed the opening sentence in the Introduction section to the following:  Perivascular spaces (PVS) pleural so that “are” is now the correct verb usage for this and other sentences to follow.

LINE 92: Furthermore, recent studies reported that EPVS are also associated to a genetic predisposition to Alzheimer’s Disease (AD), providing evidence that the biological pathways affecting AD may also influence EPVS (see, for reference: Ciampa, I.; Operto, G.; Falcon, C.; Minguillon, C.; Castro de Moura, M.; Piñeyro, D.; Esteller, M.; Molinuevo, J.L.; Guigó, R.; Navarro, A.; Gispert, J.D.; Vilor-Tejedor, N.; for the ALFA Study. Genetic Predisposition to Alzheimer’s Disease Is Associated with Enlargement of Perivascular Spaces in Centrum Semiovale Region. Genes 202112, 825. https://doi.org/10.3390/genes12060825).  Please note that this comment and new reference were added as reference 7 with the references changed accordingly in lines [113-115]

LINE 95: In this perspective, microscopy plays a crucial role, being a fundamental tool in clinical setting. If light microscopy (LM) is an essential tool for describing the most significant morphological changes of tissues, transmission electron microscopy (TEM) is a powerful instrument which provides ultrastructural evidence of tissues, to understand the physiological and pathological dynamics of a tissue (see, for reference: Torge, D.; Bernardi, S.; Arcangeli, M.; Bianchi, S. Histopathological Features of SARS-CoV-2 in Extrapulmonary Organ Infection: A Systematic Review of Literature. Pathogens 202211, 867. https://doi.org/10.3390/pathogens11080867). This is the major concern of this manuscript: the authors should discuss the importance of microscopy in clinical setting.  Authors are extremely grateful for both of these above recommendations and wish to thank reviewer number 2 as we feel this will further support the use of TEM and microscopy in general, to better understand PVS and their enlargement.  Please note the following insertion: “In this perspective, microscopy plays a crucial role, being a fundamental tool in the clinical setting. Light microscopy is an essential tool for describing the significant morphological changes of tissues and TEM is a powerful instrument, which provides ultrastructural evidence of tissue remodeling, to understand the physiological and pathological dynamics of PVS development and remodeling” in lines 115-119.

LINE 120: Microbleeds, intracerebral hemorrhages… Authors have now added intracerebral hemorrhages…

LINE 146: Initially, the response to injury wound healing mechanisms is protective. It seems that the verb “are” does not agree with the subject.  This has now been changed  to … wound healing mechanisms is

The main topic is interesting, and certainly of great clinical impact. As regards the originality and strengths of this manuscript, this is a significant contribute to the ongoing research on this topic, as it extends the research field on the EPVS association with basal ganglia (BG), centrum semiovale (CSO), lacunes, WMH, and SVD; brain endothelial cell activation and dysfunction (BECact/dys) and BEC glycocalyx (ecGCx) shedding associated with EPVS, and SVD in section 3.; the metabolic syndrome (MetS), SVD, and PVS. Overall, the contents are rich, and the authors also give their deep insight for some works.

There is a specific and detailed explanation for the evidence mentioned in this study: this is particularly significant, since the manuscript relies on a multitude of studies, to derive its conclusions.

The conclusion of this manuscript is perfectly in line with the main purpose of the paper: the authors have designed and conducted the study properly. As regards the conclusions, they are well written and present an adequate balance between the description of previous findings and the results presented by the authors.

In conclusion, this manuscript is densely presented and well organized, based on well-synthetized evidence. The authors were lucid in their style of writing, making it easy to read and understand the message, portrayed in the manuscript. Besides, the methodology design was appropriately implemented within the study. However, many of the topics are very concisely covered. This manuscript provided a comprehensive analysis of current knowledge in this field. Moreover, this research has futuristic importance and could be potential for future research. However, major concerns of this manuscript are with the introductive section: for these reasons, I have major comments for this section, for improvement before acceptance for publication. The article is accurate and provides relevant information on the topic and I have some major points to make, that may help to improve the quality of the current manuscript and maximize its scientific impact. I would accept this manuscript if the comments are addressed properly.  The authors wish to kindly thank reviewer number 2 for these kind comments and appropriate recommendations to improve the manuscript.  Thank you so very much for your precious time and knowledge for making this manuscript better for our readers.  We have made the changes that you have recommended  see lined manuscript with blue lettering and also reference changes in blue lettering.

Reviewer # 2

The aim of this manuscript is to discuss the EPVS association with basal ganglia (BG), centrum semiovale (CSO), lacunes, WMH, and SVD; brain endothelial cell activation and dysfunction (BECact/dys) and BEC glycocalyx (ecGCx) shedding associated with EPVS, and SVD in section 3.; the metabolic syndrome (MetS), SVD, and PVS.

This manuscript shows rich content, providing a deep insight for some works: the study is within the journal’s scope, and I found it to be well-written, providing sufficient information. Even if the manuscript provides an organic overview, with a densely organized structure and based on well-synthetized evidence, there are some suggestions necessary to make the article complete and fully readable. For these reasons, the manuscript requires major changes.

Please find below an enumerated list of comments on my review of the manuscript:

INTRODUCTION:

The authors should provide a list of the abbreviations, used in the manuscript, before the introductive section. Authors wish to thank reviewer number 2 for these suggestions and we agree that adding abbreviations should be done. However, we just heard from Ms Snezana Vidakovic (Section Managing Editor) that we were to place the list of abbreviations after the conclusion per her directions so we have done that.  Please see lines [511-526]

LINE 42: It seem that the verb “are” does not agree with the subject. Please note that authors have now changed the opening sentence in the Introduction section to the following:  Perivascular spaces (PVS) pleural so that “are” is now the correct verb usage for this and other sentences to follow.

LINE 92: Furthermore, recent studies reported that EPVS are also associated to a genetic predisposition to Alzheimer’s Disease (AD), providing evidence that the biological pathways affecting AD may also influence EPVS (see, for reference: Ciampa, I.; Operto, G.; Falcon, C.; Minguillon, C.; Castro de Moura, M.; Piñeyro, D.; Esteller, M.; Molinuevo, J.L.; Guigó, R.; Navarro, A.; Gispert, J.D.; Vilor-Tejedor, N.; for the ALFA Study. Genetic Predisposition to Alzheimer’s Disease Is Associated with Enlargement of Perivascular Spaces in Centrum Semiovale Region. Genes 202112, 825. https://doi.org/10.3390/genes12060825).  Please note that this comment and new reference were added as reference 7 with the references changed accordingly in lines [113-115]

LINE 95: In this perspective, microscopy plays a crucial role, being a fundamental tool in clinical setting. If light microscopy (LM) is an essential tool for describing the most significant morphological changes of tissues, transmission electron microscopy (TEM) is a powerful instrument which provides ultrastructural evidence of tissues, to understand the physiological and pathological dynamics of a tissue (see, for reference: Torge, D.; Bernardi, S.; Arcangeli, M.; Bianchi, S. Histopathological Features of SARS-CoV-2 in Extrapulmonary Organ Infection: A Systematic Review of Literature. Pathogens 202211, 867. https://doi.org/10.3390/pathogens11080867). This is the major concern of this manuscript: the authors should discuss the importance of microscopy in clinical setting.  Authors are extremely grateful for both of these above recommendations and wish to thank reviewer number 2 as we feel this will further support the use of TEM and microscopy in general, to better understand PVS and their enlargement.  Please note the following insertion: “In this perspective, microscopy plays a crucial role, being a fundamental tool in the clinical setting. Light microscopy is an essential tool for describing the significant morphological changes of tissues and TEM is a powerful instrument, which provides ultrastructural evidence of tissue remodeling, to understand the physiological and pathological dynamics of PVS development and remodeling” in lines 115-119.

LINE 120: Microbleeds, intracerebral hemorrhages… Authors have now added intracerebral hemorrhages…

LINE 146: Initially, the response to injury wound healing mechanisms is protective. It seems that the verb “are” does not agree with the subject.  This has now been changed  to … wound healing mechanisms is

The main topic is interesting, and certainly of great clinical impact. As regards the originality and strengths of this manuscript, this is a significant contribute to the ongoing research on this topic, as it extends the research field on the EPVS association with basal ganglia (BG), centrum semiovale (CSO), lacunes, WMH, and SVD; brain endothelial cell activation and dysfunction (BECact/dys) and BEC glycocalyx (ecGCx) shedding associated with EPVS, and SVD in section 3.; the metabolic syndrome (MetS), SVD, and PVS. Overall, the contents are rich, and the authors also give their deep insight for some works.

There is a specific and detailed explanation for the evidence mentioned in this study: this is particularly significant, since the manuscript relies on a multitude of studies, to derive its conclusions.

The conclusion of this manuscript is perfectly in line with the main purpose of the paper: the authors have designed and conducted the study properly. As regards the conclusions, they are well written and present an adequate balance between the description of previous findings and the results presented by the authors.

In conclusion, this manuscript is densely presented and well organized, based on well-synthetized evidence. The authors were lucid in their style of writing, making it easy to read and understand the message, portrayed in the manuscript. Besides, the methodology design was appropriately implemented within the study. However, many of the topics are very concisely covered. This manuscript provided a comprehensive analysis of current knowledge in this field. Moreover, this research has futuristic importance and could be potential for future research. However, major concerns of this manuscript are with the introductive section: for these reasons, I have major comments for this section, for improvement before acceptance for publication. The article is accurate and provides relevant information on the topic and I have some major points to make, that may help to improve the quality of the current manuscript and maximize its scientific impact. I would accept this manuscript if the comments are addressed properly.  The authors wish to kindly thank reviewer number 2 for these kind comments and appropriate recommendations to improve the manuscript.  Thank you so very much for your precious time and knowledge for making this manuscript better for our readers.  We have made the changes that you have recommended  see lined manuscript with blue lettering and also reference changes in blue lettering.

Reviewer # 2

The aim of this manuscript is to discuss the EPVS association with basal ganglia (BG), centrum semiovale (CSO), lacunes, WMH, and SVD; brain endothelial cell activation and dysfunction (BECact/dys) and BEC glycocalyx (ecGCx) shedding associated with EPVS, and SVD in section 3.; the metabolic syndrome (MetS), SVD, and PVS.

This manuscript shows rich content, providing a deep insight for some works: the study is within the journal’s scope, and I found it to be well-written, providing sufficient information. Even if the manuscript provides an organic overview, with a densely organized structure and based on well-synthetized evidence, there are some suggestions necessary to make the article complete and fully readable. For these reasons, the manuscript requires major changes.

Please find below an enumerated list of comments on my review of the manuscript:

INTRODUCTION:

The authors should provide a list of the abbreviations, used in the manuscript, before the introductive section. Authors wish to thank reviewer number 2 for these suggestions and we agree that adding abbreviations should be done. However, we just heard from Ms Snezana Vidakovic (Section Managing Editor) that we were to place the list of abbreviations after the conclusion per her directions so we have done that.  Please see lines [511-526]

LINE 42: It seem that the verb “are” does not agree with the subject. Please note that authors have now changed the opening sentence in the Introduction section to the following:  Perivascular spaces (PVS) pleural so that “are” is now the correct verb usage for this and other sentences to follow.

LINE 92: Furthermore, recent studies reported that EPVS are also associated to a genetic predisposition to Alzheimer’s Disease (AD), providing evidence that the biological pathways affecting AD may also influence EPVS (see, for reference: Ciampa, I.; Operto, G.; Falcon, C.; Minguillon, C.; Castro de Moura, M.; Piñeyro, D.; Esteller, M.; Molinuevo, J.L.; Guigó, R.; Navarro, A.; Gispert, J.D.; Vilor-Tejedor, N.; for the ALFA Study. Genetic Predisposition to Alzheimer’s Disease Is Associated with Enlargement of Perivascular Spaces in Centrum Semiovale Region. Genes 202112, 825. https://doi.org/10.3390/genes12060825).  Please note that this comment and new reference were added as reference 7 with the references changed accordingly in lines [113-115]

LINE 95: In this perspective, microscopy plays a crucial role, being a fundamental tool in clinical setting. If light microscopy (LM) is an essential tool for describing the most significant morphological changes of tissues, transmission electron microscopy (TEM) is a powerful instrument which provides ultrastructural evidence of tissues, to understand the physiological and pathological dynamics of a tissue (see, for reference: Torge, D.; Bernardi, S.; Arcangeli, M.; Bianchi, S. Histopathological Features of SARS-CoV-2 in Extrapulmonary Organ Infection: A Systematic Review of Literature. Pathogens 202211, 867. https://doi.org/10.3390/pathogens11080867). This is the major concern of this manuscript: the authors should discuss the importance of microscopy in clinical setting.  Authors are extremely grateful for both of these above recommendations and wish to thank reviewer number 2 as we feel this will further support the use of TEM and microscopy in general, to better understand PVS and their enlargement.  Please note the following insertion: “In this perspective, microscopy plays a crucial role, being a fundamental tool in the clinical setting. Light microscopy is an essential tool for describing the significant morphological changes of tissues and TEM is a powerful instrument, which provides ultrastructural evidence of tissue remodeling, to understand the physiological and pathological dynamics of PVS development and remodeling” in lines 115-119.

LINE 120: Microbleeds, intracerebral hemorrhages… Authors have now added intracerebral hemorrhages…

LINE 146: Initially, the response to injury wound healing mechanisms is protective. It seems that the verb “are” does not agree with the subject.  This has now been changed  to … wound healing mechanisms is

The main topic is interesting, and certainly of great clinical impact. As regards the originality and strengths of this manuscript, this is a significant contribute to the ongoing research on this topic, as it extends the research field on the EPVS association with basal ganglia (BG), centrum semiovale (CSO), lacunes, WMH, and SVD; brain endothelial cell activation and dysfunction (BECact/dys) and BEC glycocalyx (ecGCx) shedding associated with EPVS, and SVD in section 3.; the metabolic syndrome (MetS), SVD, and PVS. Overall, the contents are rich, and the authors also give their deep insight for some works.

There is a specific and detailed explanation for the evidence mentioned in this study: this is particularly significant, since the manuscript relies on a multitude of studies, to derive its conclusions.

The conclusion of this manuscript is perfectly in line with the main purpose of the paper: the authors have designed and conducted the study properly. As regards the conclusions, they are well written and present an adequate balance between the description of previous findings and the results presented by the authors.

In conclusion, this manuscript is densely presented and well organized, based on well-synthetized evidence. The authors were lucid in their style of writing, making it easy to read and understand the message, portrayed in the manuscript. Besides, the methodology design was appropriately implemented within the study. However, many of the topics are very concisely covered. This manuscript provided a comprehensive analysis of current knowledge in this field. Moreover, this research has futuristic importance and could be potential for future research. However, major concerns of this manuscript are with the introductive section: for these reasons, I have major comments for this section, for improvement before acceptance for publication. The article is accurate and provides relevant information on the topic and I have some major points to make, that may help to improve the quality of the current manuscript and maximize its scientific impact. I would accept this manuscript if the comments are addressed properly.  The authors wish to kindly thank reviewer number 2 for these kind comments and appropriate recommendations to improve the manuscript.  Thank you so very much for your precious time and knowledge for making this manuscript better for our readers.  We have made the changes that you have recommended  see lined manuscript with blue lettering and also reference changes in blue lettering.

Reviewer # 2

The aim of this manuscript is to discuss the EPVS association with basal ganglia (BG), centrum semiovale (CSO), lacunes, WMH, and SVD; brain endothelial cell activation and dysfunction (BECact/dys) and BEC glycocalyx (ecGCx) shedding associated with EPVS, and SVD in section 3.; the metabolic syndrome (MetS), SVD, and PVS.

This manuscript shows rich content, providing a deep insight for some works: the study is within the journal’s scope, and I found it to be well-written, providing sufficient information. Even if the manuscript provides an organic overview, with a densely organized structure and based on well-synthetized evidence, there are some suggestions necessary to make the article complete and fully readable. For these reasons, the manuscript requires major changes.

Please find below an enumerated list of comments on my review of the manuscript:

INTRODUCTION:

The authors should provide a list of the abbreviations, used in the manuscript, before the introductive section. Authors wish to thank reviewer number 2 for these suggestions and we agree that adding abbreviations should be done. However, we just heard from Ms Snezana Vidakovic (Section Managing Editor) that we were to place the list of abbreviations after the conclusion per her directions so we have done that.  Please see lines [511-526]

LINE 42: It seem that the verb “are” does not agree with the subject. Please note that authors have now changed the opening sentence in the Introduction section to the following:  Perivascular spaces (PVS) pleural so that “are” is now the correct verb usage for this and other sentences to follow.

LINE 92: Furthermore, recent studies reported that EPVS are also associated to a genetic predisposition to Alzheimer’s Disease (AD), providing evidence that the biological pathways affecting AD may also influence EPVS (see, for reference: Ciampa, I.; Operto, G.; Falcon, C.; Minguillon, C.; Castro de Moura, M.; Piñeyro, D.; Esteller, M.; Molinuevo, J.L.; Guigó, R.; Navarro, A.; Gispert, J.D.; Vilor-Tejedor, N.; for the ALFA Study. Genetic Predisposition to Alzheimer’s Disease Is Associated with Enlargement of Perivascular Spaces in Centrum Semiovale Region. Genes 202112, 825. https://doi.org/10.3390/genes12060825).  Please note that this comment and new reference were added as reference 7 with the references changed accordingly in lines [113-115]

LINE 95: In this perspective, microscopy plays a crucial role, being a fundamental tool in clinical setting. If light microscopy (LM) is an essential tool for describing the most significant morphological changes of tissues, transmission electron microscopy (TEM) is a powerful instrument which provides ultrastructural evidence of tissues, to understand the physiological and pathological dynamics of a tissue (see, for reference: Torge, D.; Bernardi, S.; Arcangeli, M.; Bianchi, S. Histopathological Features of SARS-CoV-2 in Extrapulmonary Organ Infection: A Systematic Review of Literature. Pathogens 202211, 867. https://doi.org/10.3390/pathogens11080867). This is the major concern of this manuscript: the authors should discuss the importance of microscopy in clinical setting.  Authors are extremely grateful for both of these above recommendations and wish to thank reviewer number 2 as we feel this will further support the use of TEM and microscopy in general, to better understand PVS and their enlargement.  Please note the following insertion: “In this perspective, microscopy plays a crucial role, being a fundamental tool in the clinical setting. Light microscopy is an essential tool for describing the significant morphological changes of tissues and TEM is a powerful instrument, which provides ultrastructural evidence of tissue remodeling, to understand the physiological and pathological dynamics of PVS development and remodeling” in lines 115-119.

LINE 120: Microbleeds, intracerebral hemorrhages… Authors have now added intracerebral hemorrhages…

LINE 146: Initially, the response to injury wound healing mechanisms is protective. It seems that the verb “are” does not agree with the subject.  This has now been changed  to … wound healing mechanisms is

The main topic is interesting, and certainly of great clinical impact. As regards the originality and strengths of this manuscript, this is a significant contribute to the ongoing research on this topic, as it extends the research field on the EPVS association with basal ganglia (BG), centrum semiovale (CSO), lacunes, WMH, and SVD; brain endothelial cell activation and dysfunction (BECact/dys) and BEC glycocalyx (ecGCx) shedding associated with EPVS, and SVD in section 3.; the metabolic syndrome (MetS), SVD, and PVS. Overall, the contents are rich, and the authors also give their deep insight for some works.

There is a specific and detailed explanation for the evidence mentioned in this study: this is particularly significant, since the manuscript relies on a multitude of studies, to derive its conclusions.

The conclusion of this manuscript is perfectly in line with the main purpose of the paper: the authors have designed and conducted the study properly. As regards the conclusions, they are well written and present an adequate balance between the description of previous findings and the results presented by the authors.

In conclusion, this manuscript is densely presented and well organized, based on well-synthetized evidence. The authors were lucid in their style of writing, making it easy to read and understand the message, portrayed in the manuscript. Besides, the methodology design was appropriately implemented within the study. However, many of the topics are very concisely covered. This manuscript provided a comprehensive analysis of current knowledge in this field. Moreover, this research has futuristic importance and could be potential for future research. However, major concerns of this manuscript are with the introductive section: for these reasons, I have major comments for this section, for improvement before acceptance for publication. The article is accurate and provides relevant information on the topic and I have some major points to make, that may help to improve the quality of the current manuscript and maximize its scientific impact. I would accept this manuscript if the comments are addressed properly.  The authors wish to kindly thank reviewer number 2 for these kind comments and appropriate recommendations to improve the manuscript.  Thank you so very much for your precious time and knowledge for making this manuscript better for our readers.  We have made the changes that you have recommended  see lined manuscript with blue lettering and also reference changes in blue lettering.

Reviewer # 2

The aim of this manuscript is to discuss the EPVS association with basal ganglia (BG), centrum semiovale (CSO), lacunes, WMH, and SVD; brain endothelial cell activation and dysfunction (BECact/dys) and BEC glycocalyx (ecGCx) shedding associated with EPVS, and SVD in section 3.; the metabolic syndrome (MetS), SVD, and PVS.

This manuscript shows rich content, providing a deep insight for some works: the study is within the journal’s scope, and I found it to be well-written, providing sufficient information. Even if the manuscript provides an organic overview, with a densely organized structure and based on well-synthetized evidence, there are some suggestions necessary to make the article complete and fully readable. For these reasons, the manuscript requires major changes.

Please find below an enumerated list of comments on my review of the manuscript:

INTRODUCTION:

The authors should provide a list of the abbreviations, used in the manuscript, before the introductive section. Authors wish to thank reviewer number 2 for these suggestions and we agree that adding abbreviations should be done. However, we just heard from Ms Snezana Vidakovic (Section Managing Editor) that we were to place the list of abbreviations after the conclusion per her directions so we have done that.  Please see lines [511-526]

LINE 42: It seem that the verb “are” does not agree with the subject. Please note that authors have now changed the opening sentence in the Introduction section to the following:  Perivascular spaces (PVS) pleural so that “are” is now the correct verb usage for this and other sentences to follow.

LINE 92: Furthermore, recent studies reported that EPVS are also associated to a genetic predisposition to Alzheimer’s Disease (AD), providing evidence that the biological pathways affecting AD may also influence EPVS (see, for reference: Ciampa, I.; Operto, G.; Falcon, C.; Minguillon, C.; Castro de Moura, M.; Piñeyro, D.; Esteller, M.; Molinuevo, J.L.; Guigó, R.; Navarro, A.; Gispert, J.D.; Vilor-Tejedor, N.; for the ALFA Study. Genetic Predisposition to Alzheimer’s Disease Is Associated with Enlargement of Perivascular Spaces in Centrum Semiovale Region. Genes 202112, 825. https://doi.org/10.3390/genes12060825).  Please note that this comment and new reference were added as reference 7 with the references changed accordingly in lines [113-115]

LINE 95: In this perspective, microscopy plays a crucial role, being a fundamental tool in clinical setting. If light microscopy (LM) is an essential tool for describing the most significant morphological changes of tissues, transmission electron microscopy (TEM) is a powerful instrument which provides ultrastructural evidence of tissues, to understand the physiological and pathological dynamics of a tissue (see, for reference: Torge, D.; Bernardi, S.; Arcangeli, M.; Bianchi, S. Histopathological Features of SARS-CoV-2 in Extrapulmonary Organ Infection: A Systematic Review of Literature. Pathogens 202211, 867. https://doi.org/10.3390/pathogens11080867). This is the major concern of this manuscript: the authors should discuss the importance of microscopy in clinical setting.  Authors are extremely grateful for both of these above recommendations and wish to thank reviewer number 2 as we feel this will further support the use of TEM and microscopy in general, to better understand PVS and their enlargement.  Please note the following insertion: “In this perspective, microscopy plays a crucial role, being a fundamental tool in the clinical setting. Light microscopy is an essential tool for describing the significant morphological changes of tissues and TEM is a powerful instrument, which provides ultrastructural evidence of tissue remodeling, to understand the physiological and pathological dynamics of PVS development and remodeling” in lines 115-119.

LINE 120: Microbleeds, intracerebral hemorrhages… Authors have now added intracerebral hemorrhages…

LINE 146: Initially, the response to injury wound healing mechanisms is protective. It seems that the verb “are” does not agree with the subject.  This has now been changed  to … wound healing mechanisms is

The main topic is interesting, and certainly of great clinical impact. As regards the originality and strengths of this manuscript, this is a significant contribute to the ongoing research on this topic, as it extends the research field on the EPVS association with basal ganglia (BG), centrum semiovale (CSO), lacunes, WMH, and SVD; brain endothelial cell activation and dysfunction (BECact/dys) and BEC glycocalyx (ecGCx) shedding associated with EPVS, and SVD in section 3.; the metabolic syndrome (MetS), SVD, and PVS. Overall, the contents are rich, and the authors also give their deep insight for some works.

There is a specific and detailed explanation for the evidence mentioned in this study: this is particularly significant, since the manuscript relies on a multitude of studies, to derive its conclusions.

The conclusion of this manuscript is perfectly in line with the main purpose of the paper: the authors have designed and conducted the study properly. As regards the conclusions, they are well written and present an adequate balance between the description of previous findings and the results presented by the authors.

In conclusion, this manuscript is densely presented and well organized, based on well-synthetized evidence. The authors were lucid in their style of writing, making it easy to read and understand the message, portrayed in the manuscript. Besides, the methodology design was appropriately implemented within the study. However, many of the topics are very concisely covered. This manuscript provided a comprehensive analysis of current knowledge in this field. Moreover, this research has futuristic importance and could be potential for future research. However, major concerns of this manuscript are with the introductive section: for these reasons, I have major comments for this section, for improvement before acceptance for publication. The article is accurate and provides relevant information on the topic and I have some major points to make, that may help to improve the quality of the current manuscript and maximize its scientific impact. I would accept this manuscript if the comments are addressed properly.  The authors wish to kindly thank reviewer number 2 for these kind comments and appropriate recommendations to improve the manuscript.  Thank you so very much for your precious time and knowledge for making this manuscript better for our readers.  We have made the changes that you have recommended  see lined manuscript with blue lettering and also reference changes in blue lettering.

Reviewer # 2

The aim of this manuscript is to discuss the EPVS association with basal ganglia (BG), centrum semiovale (CSO), lacunes, WMH, and SVD; brain endothelial cell activation and dysfunction (BECact/dys) and BEC glycocalyx (ecGCx) shedding associated with EPVS, and SVD in section 3.; the metabolic syndrome (MetS), SVD, and PVS.

This manuscript shows rich content, providing a deep insight for some works: the study is within the journal’s scope, and I found it to be well-written, providing sufficient information. Even if the manuscript provides an organic overview, with a densely organized structure and based on well-synthetized evidence, there are some suggestions necessary to make the article complete and fully readable. For these reasons, the manuscript requires major changes.

Please find below an enumerated list of comments on my review of the manuscript:

INTRODUCTION:

The authors should provide a list of the abbreviations, used in the manuscript, before the introductive section. Authors wish to thank reviewer number 2 for these suggestions and we agree that adding abbreviations should be done. However, we just heard from Ms Snezana Vidakovic (Section Managing Editor) that we were to place the list of abbreviations after the conclusion per her directions so we have done that.  Please see lines [511-526]

LINE 42: It seem that the verb “are” does not agree with the subject. Please note that authors have now changed the opening sentence in the Introduction section to the following:  Perivascular spaces (PVS) pleural so that “are” is now the correct verb usage for this and other sentences to follow.

LINE 92: Furthermore, recent studies reported that EPVS are also associated to a genetic predisposition to Alzheimer’s Disease (AD), providing evidence that the biological pathways affecting AD may also influence EPVS (see, for reference: Ciampa, I.; Operto, G.; Falcon, C.; Minguillon, C.; Castro de Moura, M.; Piñeyro, D.; Esteller, M.; Molinuevo, J.L.; Guigó, R.; Navarro, A.; Gispert, J.D.; Vilor-Tejedor, N.; for the ALFA Study. Genetic Predisposition to Alzheimer’s Disease Is Associated with Enlargement of Perivascular Spaces in Centrum Semiovale Region. Genes 202112, 825. https://doi.org/10.3390/genes12060825).  Please note that this comment and new reference were added as reference 7 with the references changed accordingly in lines [113-115]

LINE 95: In this perspective, microscopy plays a crucial role, being a fundamental tool in clinical setting. If light microscopy (LM) is an essential tool for describing the most significant morphological changes of tissues, transmission electron microscopy (TEM) is a powerful instrument which provides ultrastructural evidence of tissues, to understand the physiological and pathological dynamics of a tissue (see, for reference: Torge, D.; Bernardi, S.; Arcangeli, M.; Bianchi, S. Histopathological Features of SARS-CoV-2 in Extrapulmonary Organ Infection: A Systematic Review of Literature. Pathogens 202211, 867. https://doi.org/10.3390/pathogens11080867). This is the major concern of this manuscript: the authors should discuss the importance of microscopy in clinical setting.  Authors are extremely grateful for both of these above recommendations and wish to thank reviewer number 2 as we feel this will further support the use of TEM and microscopy in general, to better understand PVS and their enlargement.  Please note the following insertion: “In this perspective, microscopy plays a crucial role, being a fundamental tool in the clinical setting. Light microscopy is an essential tool for describing the significant morphological changes of tissues and TEM is a powerful instrument, which provides ultrastructural evidence of tissue remodeling, to understand the physiological and pathological dynamics of PVS development and remodeling” in lines 115-119.

LINE 120: Microbleeds, intracerebral hemorrhages… Authors have now added intracerebral hemorrhages…

LINE 146: Initially, the response to injury wound healing mechanisms is protective. It seems that the verb “are” does not agree with the subject.  This has now been changed  to … wound healing mechanisms is

The main topic is interesting, and certainly of great clinical impact. As regards the originality and strengths of this manuscript, this is a significant contribute to the ongoing research on this topic, as it extends the research field on the EPVS association with basal ganglia (BG), centrum semiovale (CSO), lacunes, WMH, and SVD; brain endothelial cell activation and dysfunction (BECact/dys) and BEC glycocalyx (ecGCx) shedding associated with EPVS, and SVD in section 3.; the metabolic syndrome (MetS), SVD, and PVS. Overall, the contents are rich, and the authors also give their deep insight for some works.

There is a specific and detailed explanation for the evidence mentioned in this study: this is particularly significant, since the manuscript relies on a multitude of studies, to derive its conclusions.

The conclusion of this manuscript is perfectly in line with the main purpose of the paper: the authors have designed and conducted the study properly. As regards the conclusions, they are well written and present an adequate balance between the description of previous findings and the results presented by the authors.

In conclusion, this manuscript is densely presented and well organized, based on well-synthetized evidence. The authors were lucid in their style of writing, making it easy to read and understand the message, portrayed in the manuscript. Besides, the methodology design was appropriately implemented within the study. However, many of the topics are very concisely covered. This manuscript provided a comprehensive analysis of current knowledge in this field. Moreover, this research has futuristic importance and could be potential for future research. However, major concerns of this manuscript are with the introductive section: for these reasons, I have major comments for this section, for improvement before acceptance for publication. The article is accurate and provides relevant information on the topic and I have some major points to make, that may help to improve the quality of the current manuscript and maximize its scientific impact. I would accept this manuscript if the comments are addressed properly.  The authors wish to kindly thank reviewer number 2 for these kind comments and appropriate recommendations to improve the manuscript.  Thank you so very much for your precious time and knowledge for making this manuscript better for our readers.  We have made the changes that you have recommended  see lined manuscript with blue lettering and also reference changes in blue lettering.

Reviewer # 2

The aim of this manuscript is to discuss the EPVS association with basal ganglia (BG), centrum semiovale (CSO), lacunes, WMH, and SVD; brain endothelial cell activation and dysfunction (BECact/dys) and BEC glycocalyx (ecGCx) shedding associated with EPVS, and SVD in section 3.; the metabolic syndrome (MetS), SVD, and PVS.

This manuscript shows rich content, providing a deep insight for some works: the study is within the journal’s scope, and I found it to be well-written, providing sufficient information. Even if the manuscript provides an organic overview, with a densely organized structure and based on well-synthetized evidence, there are some suggestions necessary to make the article complete and fully readable. For these reasons, the manuscript requires major changes.

Please find below an enumerated list of comments on my review of the manuscript:

INTRODUCTION:

The authors should provide a list of the abbreviations, used in the manuscript, before the introductive section. Authors wish to thank reviewer number 2 for these suggestions and we agree that adding abbreviations should be done. However, we just heard from Ms Snezana Vidakovic (Section Managing Editor) that we were to place the list of abbreviations after the conclusion per her directions so we have done that.  Please see lines [511-526]

LINE 42: It seem that the verb “are” does not agree with the subject. Please note that authors have now changed the opening sentence in the Introduction section to the following:  Perivascular spaces (PVS) pleural so that “are” is now the correct verb usage for this and other sentences to follow.

LINE 92: Furthermore, recent studies reported that EPVS are also associated to a genetic predisposition to Alzheimer’s Disease (AD), providing evidence that the biological pathways affecting AD may also influence EPVS (see, for reference: Ciampa, I.; Operto, G.; Falcon, C.; Minguillon, C.; Castro de Moura, M.; Piñeyro, D.; Esteller, M.; Molinuevo, J.L.; Guigó, R.; Navarro, A.; Gispert, J.D.; Vilor-Tejedor, N.; for the ALFA Study. Genetic Predisposition to Alzheimer’s Disease Is Associated with Enlargement of Perivascular Spaces in Centrum Semiovale Region. Genes 202112, 825. https://doi.org/10.3390/genes12060825).  Please note that this comment and new reference were added as reference 7 with the references changed accordingly in lines [113-115]

LINE 95: In this perspective, microscopy plays a crucial role, being a fundamental tool in clinical setting. If light microscopy (LM) is an essential tool for describing the most significant morphological changes of tissues, transmission electron microscopy (TEM) is a powerful instrument which provides ultrastructural evidence of tissues, to understand the physiological and pathological dynamics of a tissue (see, for reference: Torge, D.; Bernardi, S.; Arcangeli, M.; Bianchi, S. Histopathological Features of SARS-CoV-2 in Extrapulmonary Organ Infection: A Systematic Review of Literature. Pathogens 202211, 867. https://doi.org/10.3390/pathogens11080867). This is the major concern of this manuscript: the authors should discuss the importance of microscopy in clinical setting.  Authors are extremely grateful for both of these above recommendations and wish to thank reviewer number 2 as we feel this will further support the use of TEM and microscopy in general, to better understand PVS and their enlargement.  Please note the following insertion: “In this perspective, microscopy plays a crucial role, being a fundamental tool in the clinical setting. Light microscopy is an essential tool for describing the significant morphological changes of tissues and TEM is a powerful instrument, which provides ultrastructural evidence of tissue remodeling, to understand the physiological and pathological dynamics of PVS development and remodeling” in lines 115-119.

LINE 120: Microbleeds, intracerebral hemorrhages… Authors have now added intracerebral hemorrhages…

LINE 146: Initially, the response to injury wound healing mechanisms is protective. It seems that the verb “are” does not agree with the subject.  This has now been changed  to … wound healing mechanisms is

The main topic is interesting, and certainly of great clinical impact. As regards the originality and strengths of this manuscript, this is a significant contribute to the ongoing research on this topic, as it extends the research field on the EPVS association with basal ganglia (BG), centrum semiovale (CSO), lacunes, WMH, and SVD; brain endothelial cell activation and dysfunction (BECact/dys) and BEC glycocalyx (ecGCx) shedding associated with EPVS, and SVD in section 3.; the metabolic syndrome (MetS), SVD, and PVS. Overall, the contents are rich, and the authors also give their deep insight for some works.

There is a specific and detailed explanation for the evidence mentioned in this study: this is particularly significant, since the manuscript relies on a multitude of studies, to derive its conclusions.

The conclusion of this manuscript is perfectly in line with the main purpose of the paper: the authors have designed and conducted the study properly. As regards the conclusions, they are well written and present an adequate balance between the description of previous findings and the results presented by the authors.

In conclusion, this manuscript is densely presented and well organized, based on well-synthetized evidence. The authors were lucid in their style of writing, making it easy to read and understand the message, portrayed in the manuscript. Besides, the methodology design was appropriately implemented within the study. However, many of the topics are very concisely covered. This manuscript provided a comprehensive analysis of current knowledge in this field. Moreover, this research has futuristic importance and could be potential for future research. However, major concerns of this manuscript are with the introductive section: for these reasons, I have major comments for this section, for improvement before acceptance for publication. The article is accurate and provides relevant information on the topic and I have some major points to make, that may help to improve the quality of the current manuscript and maximize its scientific impact. I would accept this manuscript if the comments are addressed properly.  The authors wish to kindly thank reviewer number 2 for these kind comments and appropriate recommendations to improve the manuscript.  Thank you so very much for your precious time and knowledge for making this manuscript better for our readers.  We have made the changes that you have recommended  see lined manuscript with blue lettering and also reference changes in blue lettering.

Reviewer # 2

The aim of this manuscript is to discuss the EPVS association with basal ganglia (BG), centrum semiovale (CSO), lacunes, WMH, and SVD; brain endothelial cell activation and dysfunction (BECact/dys) and BEC glycocalyx (ecGCx) shedding associated with EPVS, and SVD in section 3.; the metabolic syndrome (MetS), SVD, and PVS.

This manuscript shows rich content, providing a deep insight for some works: the study is within the journal’s scope, and I found it to be well-written, providing sufficient information. Even if the manuscript provides an organic overview, with a densely organized structure and based on well-synthetized evidence, there are some suggestions necessary to make the article complete and fully readable. For these reasons, the manuscript requires major changes.

Please find below an enumerated list of comments on my review of the manuscript:

INTRODUCTION:

The authors should provide a list of the abbreviations, used in the manuscript, before the introductive section. Authors wish to thank reviewer number 2 for these suggestions and we agree that adding abbreviations should be done. However, we just heard from Ms Snezana Vidakovic (Section Managing Editor) that we were to place the list of abbreviations after the conclusion per her directions so we have done that.  Please see lines [511-526]

LINE 42: It seem that the verb “are” does not agree with the subject. Please note that authors have now changed the opening sentence in the Introduction section to the following:  Perivascular spaces (PVS) pleural so that “are” is now the correct verb usage for this and other sentences to follow.

LINE 92: Furthermore, recent studies reported that EPVS are also associated to a genetic predisposition to Alzheimer’s Disease (AD), providing evidence that the biological pathways affecting AD may also influence EPVS (see, for reference: Ciampa, I.; Operto, G.; Falcon, C.; Minguillon, C.; Castro de Moura, M.; Piñeyro, D.; Esteller, M.; Molinuevo, J.L.; Guigó, R.; Navarro, A.; Gispert, J.D.; Vilor-Tejedor, N.; for the ALFA Study. Genetic Predisposition to Alzheimer’s Disease Is Associated with Enlargement of Perivascular Spaces in Centrum Semiovale Region. Genes 202112, 825. https://doi.org/10.3390/genes12060825).  Please note that this comment and new reference were added as reference 7 with the references changed accordingly in lines [113-115]

LINE 95: In this perspective, microscopy plays a crucial role, being a fundamental tool in clinical setting. If light microscopy (LM) is an essential tool for describing the most significant morphological changes of tissues, transmission electron microscopy (TEM) is a powerful instrument which provides ultrastructural evidence of tissues, to understand the physiological and pathological dynamics of a tissue (see, for reference: Torge, D.; Bernardi, S.; Arcangeli, M.; Bianchi, S. Histopathological Features of SARS-CoV-2 in Extrapulmonary Organ Infection: A Systematic Review of Literature. Pathogens 202211, 867. https://doi.org/10.3390/pathogens11080867). This is the major concern of this manuscript: the authors should discuss the importance of microscopy in clinical setting.  Authors are extremely grateful for both of these above recommendations and wish to thank reviewer number 2 as we feel this will further support the use of TEM and microscopy in general, to better understand PVS and their enlargement.  Please note the following insertion: “In this perspective, microscopy plays a crucial role, being a fundamental tool in the clinical setting. Light microscopy is an essential tool for describing the significant morphological changes of tissues and TEM is a powerful instrument, which provides ultrastructural evidence of tissue remodeling, to understand the physiological and pathological dynamics of PVS development and remodeling” in lines 115-119.

LINE 120: Microbleeds, intracerebral hemorrhages… Authors have now added intracerebral hemorrhages…

LINE 146: Initially, the response to injury wound healing mechanisms is protective. It seems that the verb “are” does not agree with the subject.  This has now been changed  to … wound healing mechanisms is

The main topic is interesting, and certainly of great clinical impact. As regards the originality and strengths of this manuscript, this is a significant contribute to the ongoing research on this topic, as it extends the research field on the EPVS association with basal ganglia (BG), centrum semiovale (CSO), lacunes, WMH, and SVD; brain endothelial cell activation and dysfunction (BECact/dys) and BEC glycocalyx (ecGCx) shedding associated with EPVS, and SVD in section 3.; the metabolic syndrome (MetS), SVD, and PVS. Overall, the contents are rich, and the authors also give their deep insight for some works.

There is a specific and detailed explanation for the evidence mentioned in this study: this is particularly significant, since the manuscript relies on a multitude of studies, to derive its conclusions.

The conclusion of this manuscript is perfectly in line with the main purpose of the paper: the authors have designed and conducted the study properly. As regards the conclusions, they are well written and present an adequate balance between the description of previous findings and the results presented by the authors.

In conclusion, this manuscript is densely presented and well organized, based on well-synthetized evidence. The authors were lucid in their style of writing, making it easy to read and understand the message, portrayed in the manuscript. Besides, the methodology design was appropriately implemented within the study. However, many of the topics are very concisely covered. This manuscript provided a comprehensive analysis of current knowledge in this field. Moreover, this research has futuristic importance and could be potential for future research. However, major concerns of this manuscript are with the introductive section: for these reasons, I have major comments for this section, for improvement before acceptance for publication. The article is accurate and provides relevant information on the topic and I have some major points to make, that may help to improve the quality of the current manuscript and maximize its scientific impact. I would accept this manuscript if the comments are addressed properly.  The authors wish to kindly thank reviewer number 2 for these kind comments and appropriate recommendations to improve the manuscript.  Thank you so very much for your precious time and knowledge for making this manuscript better for our readers.  We have made the changes that you have recommended  see lined manuscript with blue lettering and also reference changes in blue lettering.

Reviewer # 2

The aim of this manuscript is to discuss the EPVS association with basal ganglia (BG), centrum semiovale (CSO), lacunes, WMH, and SVD; brain endothelial cell activation and dysfunction (BECact/dys) and BEC glycocalyx (ecGCx) shedding associated with EPVS, and SVD in section 3.; the metabolic syndrome (MetS), SVD, and PVS.

This manuscript shows rich content, providing a deep insight for some works: the study is within the journal’s scope, and I found it to be well-written, providing sufficient information. Even if the manuscript provides an organic overview, with a densely organized structure and based on well-synthetized evidence, there are some suggestions necessary to make the article complete and fully readable. For these reasons, the manuscript requires major changes.

Please find below an enumerated list of comments on my review of the manuscript:

INTRODUCTION:

The authors should provide a list of the abbreviations, used in the manuscript, before the introductive section. Authors wish to thank reviewer number 2 for these suggestions and we agree that adding abbreviations should be done. However, we just heard from Ms Snezana Vidakovic (Section Managing Editor) that we were to place the list of abbreviations after the conclusion per her directions so we have done that.  Please see lines [511-526]

LINE 42: It seem that the verb “are” does not agree with the subject. Please note that authors have now changed the opening sentence in the Introduction section to the following:  Perivascular spaces (PVS) pleural so that “are” is now the correct verb usage for this and other sentences to follow.

LINE 92: Furthermore, recent studies reported that EPVS are also associated to a genetic predisposition to Alzheimer’s Disease (AD), providing evidence that the biological pathways affecting AD may also influence EPVS (see, for reference: Ciampa, I.; Operto, G.; Falcon, C.; Minguillon, C.; Castro de Moura, M.; Piñeyro, D.; Esteller, M.; Molinuevo, J.L.; Guigó, R.; Navarro, A.; Gispert, J.D.; Vilor-Tejedor, N.; for the ALFA Study. Genetic Predisposition to Alzheimer’s Disease Is Associated with Enlargement of Perivascular Spaces in Centrum Semiovale Region. Genes 202112, 825. https://doi.org/10.3390/genes12060825).  Please note that this comment and new reference were added as reference 7 with the references changed accordingly in lines [113-115]

LINE 95: In this perspective, microscopy plays a crucial role, being a fundamental tool in clinical setting. If light microscopy (LM) is an essential tool for describing the most significant morphological changes of tissues, transmission electron microscopy (TEM) is a powerful instrument which provides ultrastructural evidence of tissues, to understand the physiological and pathological dynamics of a tissue (see, for reference: Torge, D.; Bernardi, S.; Arcangeli, M.; Bianchi, S. Histopathological Features of SARS-CoV-2 in Extrapulmonary Organ Infection: A Systematic Review of Literature. Pathogens 202211, 867. https://doi.org/10.3390/pathogens11080867). This is the major concern of this manuscript: the authors should discuss the importance of microscopy in clinical setting.  Authors are extremely grateful for both of these above recommendations and wish to thank reviewer number 2 as we feel this will further support the use of TEM and microscopy in general, to better understand PVS and their enlargement.  Please note the following insertion: “In this perspective, microscopy plays a crucial role, being a fundamental tool in the clinical setting. Light microscopy is an essential tool for describing the significant morphological changes of tissues and TEM is a powerful instrument, which provides ultrastructural evidence of tissue remodeling, to understand the physiological and pathological dynamics of PVS development and remodeling” in lines 115-119.

LINE 120: Microbleeds, intracerebral hemorrhages… Authors have now added intracerebral hemorrhages…

LINE 146: Initially, the response to injury wound healing mechanisms is protective. It seems that the verb “are” does not agree with the subject.  This has now been changed  to … wound healing mechanisms is

The main topic is interesting, and certainly of great clinical impact. As regards the originality and strengths of this manuscript, this is a significant contribute to the ongoing research on this topic, as it extends the research field on the EPVS association with basal ganglia (BG), centrum semiovale (CSO), lacunes, WMH, and SVD; brain endothelial cell activation and dysfunction (BECact/dys) and BEC glycocalyx (ecGCx) shedding associated with EPVS, and SVD in section 3.; the metabolic syndrome (MetS), SVD, and PVS. Overall, the contents are rich, and the authors also give their deep insight for some works.

There is a specific and detailed explanation for the evidence mentioned in this study: this is particularly significant, since the manuscript relies on a multitude of studies, to derive its conclusions.

The conclusion of this manuscript is perfectly in line with the main purpose of the paper: the authors have designed and conducted the study properly. As regards the conclusions, they are well written and present an adequate balance between the description of previous findings and the results presented by the authors.

In conclusion, this manuscript is densely presented and well organized, based on well-synthetized evidence. The authors were lucid in their style of writing, making it easy to read and understand the message, portrayed in the manuscript. Besides, the methodology design was appropriately implemented within the study. However, many of the topics are very concisely covered. This manuscript provided a comprehensive analysis of current knowledge in this field. Moreover, this research has futuristic importance and could be potential for future research. However, major concerns of this manuscript are with the introductive section: for these reasons, I have major comments for this section, for improvement before acceptance for publication. The article is accurate and provides relevant information on the topic and I have some major points to make, that may help to improve the quality of the current manuscript and maximize its scientific impact. I would accept this manuscript if the comments are addressed properly.  The authors wish to kindly thank reviewer number 2 for these kind comments and appropriate recommendations to improve the manuscript.  Thank you so very much for your precious time and knowledge for making this manuscript better for our readers.  We have made the changes that you have recommended  see lined manuscript with blue lettering and also reference changes in blue lettering.

Reviewer 3 Report

In this review, Tatyana Shulyatnikova and Melvin R Hayden discussed why EPVS are important and understanding the underlying mechanism of this maladaptation might help mitigate the aging-related diseases associated with EPVS.

Overall this review is well-written, and the illustrations are well-detailed to facilitate comprehension by its readership.

Author Response

Reviewer # 3

In this review, Tatyana Shulyatnikova and Melvin R Hayden discussed why EPVS are important and understanding the underlying mechanism of this maladaptation might help mitigate the aging-related diseases associated with EPVS.

Overall this review is well-written, and the illustrations are well-detailed to facilitate comprehension by its readership. Authors wish to thank reviewer number 3 for the kind comments.

Reviewer # 3

In this review, Tatyana Shulyatnikova and Melvin R Hayden discussed why EPVS are important and understanding the underlying mechanism of this maladaptation might help mitigate the aging-related diseases associated with EPVS.

Overall this review is well-written, and the illustrations are well-detailed to facilitate comprehension by its readership. Authors wish to thank reviewer number 3 for the kind comments.

Reviewer # 3

In this review, Tatyana Shulyatnikova and Melvin R Hayden discussed why EPVS are important and understanding the underlying mechanism of this maladaptation might help mitigate the aging-related diseases associated with EPVS.

Overall this review is well-written, and the illustrations are well-detailed to facilitate comprehension by its readership. Authors wish to thank reviewer number 3 for the kind comments.

Reviewer # 3

In this review, Tatyana Shulyatnikova and Melvin R Hayden discussed why EPVS are important and understanding the underlying mechanism of this maladaptation might help mitigate the aging-related diseases associated with EPVS.

Overall this review is well-written, and the illustrations are well-detailed to facilitate comprehension by its readership. Authors wish to thank reviewer number 3 for the kind comments.

Reviewer # 3

In this review, Tatyana Shulyatnikova and Melvin R Hayden discussed why EPVS are important and understanding the underlying mechanism of this maladaptation might help mitigate the aging-related diseases associated with EPVS.

Overall this review is well-written, and the illustrations are well-detailed to facilitate comprehension by its readership. Authors wish to thank reviewer number 3 for the kind comments.

Reviewer # 3

In this review, Tatyana Shulyatnikova and Melvin R Hayden discussed why EPVS are important and understanding the underlying mechanism of this maladaptation might help mitigate the aging-related diseases associated with EPVS.

Overall this review is well-written, and the illustrations are well-detailed to facilitate comprehension by its readership. Authors wish to thank reviewer number 3 for the kind comments.

Reviewer # 3

In this review, Tatyana Shulyatnikova and Melvin R Hayden discussed why EPVS are important and understanding the underlying mechanism of this maladaptation might help mitigate the aging-related diseases associated with EPVS.

Overall this review is well-written, and the illustrations are well-detailed to facilitate comprehension by its readership. Authors wish to thank reviewer number 3 for the kind comments.

Reviewer # 3

In this review, Tatyana Shulyatnikova and Melvin R Hayden discussed why EPVS are important and understanding the underlying mechanism of this maladaptation might help mitigate the aging-related diseases associated with EPVS.

Overall this review is well-written, and the illustrations are well-detailed to facilitate comprehension by its readership. Authors wish to thank reviewer number 3 for the kind comments.

Reviewer # 3

In this review, Tatyana Shulyatnikova and Melvin R Hayden discussed why EPVS are important and understanding the underlying mechanism of this maladaptation might help mitigate the aging-related diseases associated with EPVS.

Overall this review is well-written, and the illustrations are well-detailed to facilitate comprehension by its readership. Authors wish to thank reviewer number 3 for the kind comments.

Reviewer # 3

In this review, Tatyana Shulyatnikova and Melvin R Hayden discussed why EPVS are important and understanding the underlying mechanism of this maladaptation might help mitigate the aging-related diseases associated with EPVS.

Overall this review is well-written, and the illustrations are well-detailed to facilitate comprehension by its readership. Authors wish to thank reviewer number 3 for the kind comments.

Reviewer # 3

In this review, Tatyana Shulyatnikova and Melvin R Hayden discussed why EPVS are important and understanding the underlying mechanism of this maladaptation might help mitigate the aging-related diseases associated with EPVS.

Overall this review is well-written, and the illustrations are well-detailed to facilitate comprehension by its readership. Authors wish to thank reviewer number 3 for the kind comments.

Reviewer # 3

In this review, Tatyana Shulyatnikova and Melvin R Hayden discussed why EPVS are important and understanding the underlying mechanism of this maladaptation might help mitigate the aging-related diseases associated with EPVS.

Overall this review is well-written, and the illustrations are well-detailed to facilitate comprehension by its readership. Authors wish to thank reviewer number 3 for the kind comments.

Reviewer # 3

In this review, Tatyana Shulyatnikova and Melvin R Hayden discussed why EPVS are important and understanding the underlying mechanism of this maladaptation might help mitigate the aging-related diseases associated with EPVS.

Overall this review is well-written, and the illustrations are well-detailed to facilitate comprehension by its readership. Authors wish to thank reviewer number 3 for the kind comments.

Reviewer # 3

In this review, Tatyana Shulyatnikova and Melvin R Hayden discussed why EPVS are important and understanding the underlying mechanism of this maladaptation might help mitigate the aging-related diseases associated with EPVS.

Overall this review is well-written, and the illustrations are well-detailed to facilitate comprehension by its readership. Authors wish to thank reviewer number 3 for the kind comments.

Reviewer # 3

In this review, Tatyana Shulyatnikova and Melvin R Hayden discussed why EPVS are important and understanding the underlying mechanism of this maladaptation might help mitigate the aging-related diseases associated with EPVS.

Overall this review is well-written, and the illustrations are well-detailed to facilitate comprehension by its readership. Authors wish to thank reviewer number 3 for the kind comments.

Reviewer # 3

In this review, Tatyana Shulyatnikova and Melvin R Hayden discussed why EPVS are important and understanding the underlying mechanism of this maladaptation might help mitigate the aging-related diseases associated with EPVS.

Overall this review is well-written, and the illustrations are well-detailed to facilitate comprehension by its readership. Authors wish to thank reviewer number 3 for the kind comments.

Reviewer # 3

In this review, Tatyana Shulyatnikova and Melvin R Hayden discussed why EPVS are important and understanding the underlying mechanism of this maladaptation might help mitigate the aging-related diseases associated with EPVS.

Overall this review is well-written, and the illustrations are well-detailed to facilitate comprehension by its readership. Authors wish to thank reviewer number 3 for the kind comments.

Reviewer # 3

In this review, Tatyana Shulyatnikova and Melvin R Hayden discussed why EPVS are important and understanding the underlying mechanism of this maladaptation might help mitigate the aging-related diseases associated with EPVS.

Overall this review is well-written, and the illustrations are well-detailed to facilitate comprehension by its readership. Authors wish to thank reviewer number 3 for the kind comments.

Reviewer # 3

In this review, Tatyana Shulyatnikova and Melvin R Hayden discussed why EPVS are important and understanding the underlying mechanism of this maladaptation might help mitigate the aging-related diseases associated with EPVS.

Overall this review is well-written, and the illustrations are well-detailed to facilitate comprehension by its readership. Authors wish to thank reviewer number 3 for the kind comments.

Reviewer # 3

In this review, Tatyana Shulyatnikova and Melvin R Hayden discussed why EPVS are important and understanding the underlying mechanism of this maladaptation might help mitigate the aging-related diseases associated with EPVS.

Overall this review is well-written, and the illustrations are well-detailed to facilitate comprehension by its readership. Authors wish to thank reviewer number 3 for the kind comments.

Reviewer # 3

In this review, Tatyana Shulyatnikova and Melvin R Hayden discussed why EPVS are important and understanding the underlying mechanism of this maladaptation might help mitigate the aging-related diseases associated with EPVS.

Overall this review is well-written, and the illustrations are well-detailed to facilitate comprehension by its readership. Authors wish to thank reviewer number 3 for the kind comments.

Reviewer # 3

In this review, Tatyana Shulyatnikova and Melvin R Hayden discussed why EPVS are important and understanding the underlying mechanism of this maladaptation might help mitigate the aging-related diseases associated with EPVS.

Overall this review is well-written, and the illustrations are well-detailed to facilitate comprehension by its readership. Authors wish to thank reviewer number 3 for the kind comments.

Reviewer # 3

In this review, Tatyana Shulyatnikova and Melvin R Hayden discussed why EPVS are important and understanding the underlying mechanism of this maladaptation might help mitigate the aging-related diseases associated with EPVS.

Overall this review is well-written, and the illustrations are well-detailed to facilitate comprehension by its readership. Authors wish to thank reviewer number 3 for the kind comments.

Reviewer # 3

In this review, Tatyana Shulyatnikova and Melvin R Hayden discussed why EPVS are important and understanding the underlying mechanism of this maladaptation might help mitigate the aging-related diseases associated with EPVS.

Overall this review is well-written, and the illustrations are well-detailed to facilitate comprehension by its readership. Authors wish to thank reviewer number 3 for the kind comments.

Reviewer # 3

In this review, Tatyana Shulyatnikova and Melvin R Hayden discussed why EPVS are important and understanding the underlying mechanism of this maladaptation might help mitigate the aging-related diseases associated with EPVS.

Overall this review is well-written, and the illustrations are well-detailed to facilitate comprehension by its readership. Authors wish to thank reviewer number 3 for the kind comments.

Reviewer # 3

In this review, Tatyana Shulyatnikova and Melvin R Hayden discussed why EPVS are important and understanding the underlying mechanism of this maladaptation might help mitigate the aging-related diseases associated with EPVS.

Overall this review is well-written, and the illustrations are well-detailed to facilitate comprehension by its readership. Authors wish to thank reviewer number 3 for the kind comments.

Reviewer # 3

In this review, Tatyana Shulyatnikova and Melvin R Hayden discussed why EPVS are important and understanding the underlying mechanism of this maladaptation might help mitigate the aging-related diseases associated with EPVS.

Overall this review is well-written, and the illustrations are well-detailed to facilitate comprehension by its readership. Authors wish to thank reviewer number 3 for the kind comments.

Round 2

Reviewer 2 Report

The authors have improved this manuscript in an organic manner. I accept the manuscript for the publication.